# Advanced Two-Dimensional Heterojunction Photocatalysts of Stoichiometric and Non-Stoichiometric Bismuth Oxyhalides with Graphitic Carbon Nitride for Sustainable Energy and Environmental Applications

Kishore Sridharan [1,*], Sulakshana Shenoy [2], S. Girish Kumar [3], Chiaki Terashima [4], Akira Fujishima [4] and Sudhagar Pitchaimuthu [5,*]

1 Department of Nanoscience and Technology, University of Calicut, Thenhipalam 673635, India
2 Department of Physics, National Institute of Technology Karnataka, Mangalore 575025, India; shenoy26sulakshana@gmail.com
3 Department of Chemistry, School of Engineering & Technology, CMR University, Bangalore 562149, India; girichem@yahoo.co.in
4 Photocatalysis International Research Center, Research Institute for Science and Technology, Tokyo University of Science, 2641 Yamazaki, Noda Chiba 278-8510, Japan; terashima@rs.tus.ac.jp (C.T.); fujishima_akira@rs.tus.ac.jp (A.F.)
5 Multifunctional Photocatalyst and Coatings Group, SPECIFIC, Materials Research Centre, Faculty of Science and Engineering, Swansea University, Swansea SA1 8EN, UK
* Correspondence: sridharankishore@uoc.ac.in or sridharankishore@gmail.com (K.S.); s.pitchaimuthu@swansea.ac.uk or vedichi@gmail.com (S.P.)

**Abstract:** Semiconductor-based photocatalysis has been identified as an encouraging approach for solving the two main challenging problems, viz., remedying our polluted environment and the generation of sustainable chemical energy. Stoichiometric and non-stoichiometric bismuth oxyhalides (BiOX and $Bi_xO_yX_z$ where X = Cl, Br, and I) are a relatively new class of semiconductors that have attracted considerable interest for photocatalysis applications due to attributes, viz., high stability, suitable band structure, modifiable energy bandgap and two-dimensional layered structure capable of generating an internal electric field. Recently, the construction of heterojunction photocatalysts, especially 2D/2D systems, has convincingly drawn momentous attention practicably owing to the productive influence of having two dissimilar layered semiconductors in face-to-face contact with each other. This review has systematically summarized the recent progress on the 2D/2D heterojunction constructed between $BiOX/Bi_xO_yX_z$ with graphitic carbon nitride (g-$C_3N_4$). The band structure of individual components, various fabrication methods, different strategies developed for improving the photocatalytic performance and their applications in the degradation of various organic contaminants, hydrogen ($H_2$) evolution, carbon dioxide ($CO_2$) reduction, nitrogen ($N_2$) fixation and the organic synthesis of clean chemicals are summarized. The perspectives and plausible opportunities for developing high performance $BiOX/Bi_xO_yX_z$-g-$C_3N_4$ heterojunction photocatalysts are also discussed.

**Keywords:** 2D materials; photocatalysis; heterojunction; bismuth oxyhalides; graphitic carbon nitride

## 1. Introduction

Excessive demand for pharmaceutical, personal care, agricultural and industrial products driven by the continued growth of the world population has inevitably escalated the discharge of organic contaminants into the environment [1]. The steadily increasing concentration of organic contaminants primarily originating from pharmaceutical and personal care products in municipal wastewaters of many urban cities globally is making microorganisms resistant to drugs [2]. Undoubtedly, these organic contaminants pose a huge threat to the environment and human health as they have demonstrated severe ecological risk

for mutagenesis, teratogenesis and carcinogenicity [3]. Therefore, in addition to finding sustainable solutions to our global energy crisis and eliminating the steadily increasing $CO_2$ concentration from the environment, the removal of these organic contaminants with high chemical stability is another highly challenging task [4]. Several methods based on chemical [5] and biological [6] techniques and advanced oxidation processes [7] have been employed for the complete removal of organic contaminants from wastewater. However, almost all strategies failed to achieve complete degradation, and the search for a green, efficient and economically viable technology continued. In 1972, pioneering work reported by Fujishima and Honda revealed that UV light irradiated on the surface of a $TiO_2$ electrode generated free radicals for the decomposition of water into hydrogen and oxygen. Later on, it was revealed that the photogenerated free radicals emanating from semiconductors under UV/Visible light excitation could also cleavage the chemical bonds in the molecular organic contaminants adsorbed on their surfaces [8]. In this regard, heterogeneous semiconductor photocatalysis—categorized as another form of advanced oxidation process—has received an overwhelming research interest as a "one-step solution" for addressing the energy and environmental issues, viz., the generation of hydrogen gas through light-water splitting reaction, the reduction of $CO_2$ into hydrocarbons and to completely break down organic contaminants through redox reactions involving the radical species [9]. Despite nanostructured $TiO_2$ being a robust and chemically stable semiconductor, its wide bandgap energy (3.2 eV) demands UV light for its excitation. Since the visible light is predominant in the solar spectrum and with UV light being insignificant (just ~4%), researchers swiftly moved to utilize nanostructured semiconductors with a narrower bandgap energy (such as CdS, $Fe_2O_3$, $WO_3$, etc.) for efficiently utilizing the inexhaustible sunlight energy [10,11].

Since the discovery of graphene, semiconductors with 2D layered structures have greatly influenced the researchers to study them for applications in photocatalysis due to their unique sheet-like morphology with one-dimensionally confined electrons producing exceptional physio-chemical, optical and electronic properties [12]. In addition to the ease of fabrication, other interesting features of 2D semiconductors exclusively for photocatalytic applications are their large specific surface area with many photoactive sites and customizable thickness leading to easy adjustments to the bandgap energy and light absorption efficiency. Further, the atomically thin 2D layered morphology enables strong in-plane bond formation, facilitating easy heterostructure construction (on substrates or with other 2D semiconductors through weak van der Waals interaction) and enhancing the rate of the photocatalytic reactions due to the shortened transport path [13].

Among the various 2D semiconductors for photocatalysis applications, bismuth oxyhalides (referred to hereafter as BiOX, where X = Cl, Br and I)—a group of V-VI-VII ternary compounds with stoichiometric form—have become the prime choice for researchers owing to their nontoxicity, layered morphology, unique crystal structure, suitable band structure, variable bandgap energy and excellent chemical stability ensuring corrosion resistance in the solution medium for long term operations [14]. The stoichiometric BiOX possessing tetragonal matlockite polymorph (PbFCl-type; space group—$P4/nmm$) crystallize into layered structures consisting of patterned [X-Bi-O-Bi-X] slices stacked together by the nonbonding van der Waals interaction through the halogen atoms along the c-axis, as depicted in Figure 1. In each [X-Bi-O-Bi-X] layer, the central Bi atom is surrounded by four oxygen and four halogen atoms, generating an asymmetric decahedral geometry [15]. The open crystalline structure, indirect bandgap, strong covalent bonding combined with weak interlayer van der Waals interaction, and excellent electrical, optical and mechanical properties are the features that endow BiOX as a promising candidate for light induced redox reactions [16]. However, poor light absorption, restricted utilization and limited chemical stability are some of the shortcomings of BiOX.

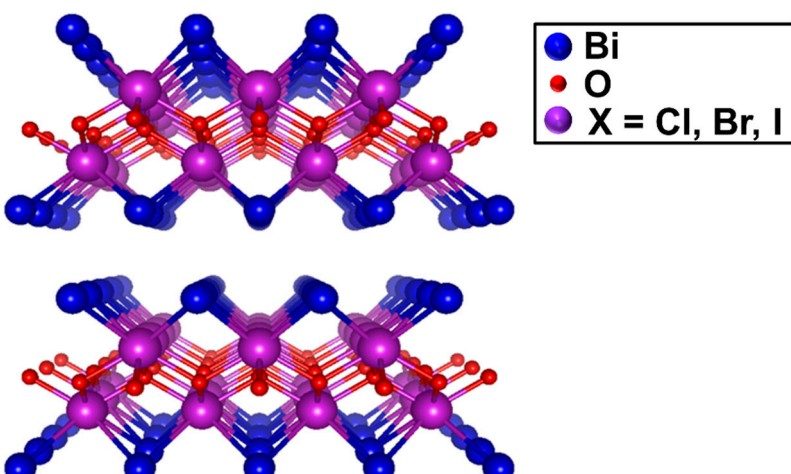

**Figure 1.** Crystal structure of the BiOX systems (space group *P*4/*nmm*, $D_{4h}$ symmetry) with stoichiometric X-Bi-O-Bi-X Bi-layers stacked along the c axis. Bismuth, oxygen, and halide ions are denoted by purple, red and blue spheres, respectively.

On the other hand, bismuth rich-bismuth oxyhalides (referred to hereafter as $Bi_xO_yX_z$) with non-stoichiometric form also have a layered structure similar to BiOX, with strong covalent bonding and weak interlayer van der Walls interactions. Generally, the band structure of a semiconductor is governed by its chemical components to a great extent. In non-stoichiometric $Bi_xO_yX_z$, the replacement of the halogen atoms in its lattice correspondingly led to modified band structure and subsequently the optical absorption edge and band redox potentials [17]. Most importantly, the negative conduction band positions of $Bi_xO_yX_z$ facilitate its widespread utilization for photocatalytic applications [18].

Graphitic carbon nitride (g-$C_3N_4$) is another exquisite 2D semiconductor that has been flourishing in the recent years for applications in photocatalysis due to its tri-s-triazine ring structure, appealing electronic band structure, medium bandgap (2.7 eV), and excellent chemical and thermal stability [19]. In addition, the earth-abundant carbon and nitrogen elements in g-$C_3N_4$ can be easily prepared via one-step polymerization of abundantly available inexpensive nitrogen-rich precursors, such as urea, thiourea, melamine, cyanamide and dicyandiamide [20,21]. Nevertheless, pristine g-$C_3N_4$ also suffers from shortcomings such as high excitation energy, low charge carrier mobility, the rapid recombination of photogenerated charge carriers, and narrow visible light absorption efficiency [22].

Thus, integrating BiOX/$Bi_xO_yX_z$ with g-$C_3N_4$ would be an ideal strategy to overcome many of the demerits associated with individual components. The 2D layered structures of both BiOX/$Bi_xO_yX_z$ and g-$C_3N_4$ conveniently promote the construction of a heterojunction and, furthermore, the favourable band energy between them can facilitate enhanced photocatalytic performance [23–29]. Several review articles on single component 2D semiconductor photocatalysts concentrating primarily on BiOX, $Bi_xO_yX_z$ and g-$C_3N_4$ have been published [15,30–58]. Nonetheless, a review article accounting the progress of heterojunction photocatalysts based on BiOX and $Bi_xO_yX_z$ with g-$C_3N_4$ is rarely reported. Since there is a consistent upsurge in the research trend on BiOX based photocatalysts as evidenced from the literature survey presented in Figure 2, a review article is needed to fill the gaps and to account the recent progress. Therefore, in this review, we have presented a summary on the band structure of $Bi_xO_yX_z$ and have furnished information on the various methods of coupling BiOX/$Bi_xO_yX$ and g-$C_3N_4$ to fabricate heterojunction photocatalysts for organic contaminant degradation, $H_2$ generation, $CO_2$ reduction, $N_2$ fixation and organic synthesis applications. Further, the various strategies for improving the performance of g-$C_3N_4$-BiOX/$Bi_xO_yX_z$ heterojunction photocatalysts, viz., the creation of defects, the role of facets, integration with other semiconductors, metals and carbon materials are discussed.

Additionally, the future prospects of BiOX/Bi$_x$O$_y$X$_z$-g-C$_3$N$_4$ heterojunction photocatalysts for broader energy and environmental applications are deliberated.

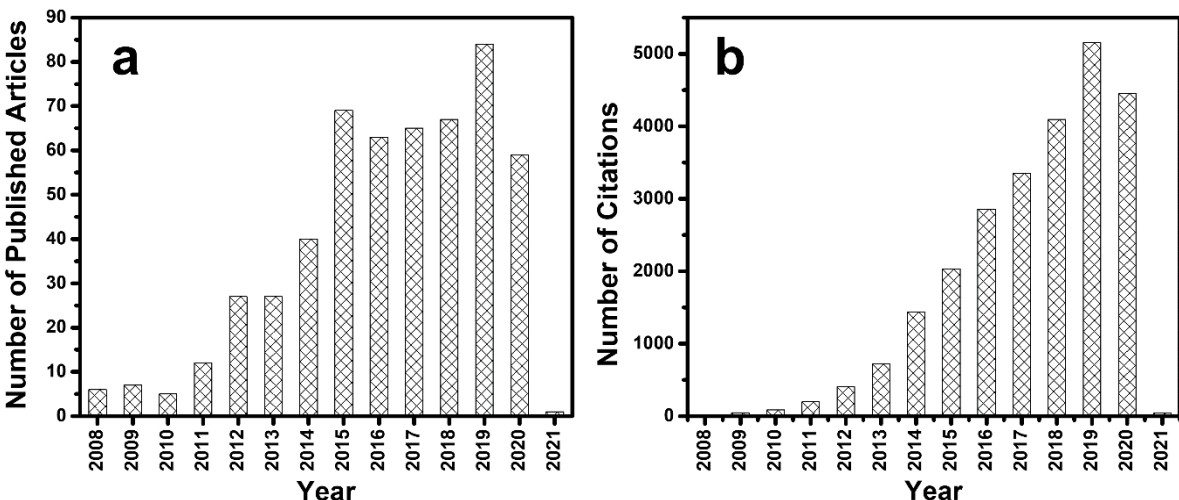

**Figure 2.** (**a**) Number of articles published and (**b**) number of citations since 2008 with topic keywords "BiOX" and "photocatal*" adapted from the Web of Science, dated 17 November 2020.

## 2. Electronic Band Structure of BiOX, Bi$_x$O$_y$X$_z$ and g-C$_3$N$_4$

The band structure of the material is the crucial parameter that dictates the light absorption capacity, charge carrier dynamics and generation of free radicals. In the case of BiOX, O 2p and X $n$p states ($n$ = 3, 4 and 5 for X = Cl, Br, and I, respectively) constitute the valance band minimum (VBM), while the conduction band maximum (CBM) is derived from Bi 6s and the Bi 6p states. The largely dispersed Bi 6s orbital facilitates the mobility of photoinduced holes in the VB (valence band) and is beneficial for the oxidation reaction. The energy bandgap values and the redox potentials of BiOX are vastly related to the atomic numbers of X and the composition of the layered structure [59]. Therefore, the optical absorption in BiOX can be tailored via varying the halogen species or Bi/X ratios. Increasing the atomic number of X leads to a change in the colour and bandgap energy of BiOX from white (BiOCl, 3.2 eV) to yellow (BiOBr, 2.7 eV) and red (BiOI, 1.7 eV), thus maximizing their light absorption capacities [60]. The open crystalline structure has a layered Sillen–Aurivillius related oxide structure composed of [Bi$_2$O$_2$] layers sandwiched between two slabs of [X] ions, and the electrostatic potential difference between the slabs generates a static internal electric field (IEF). The static IEF in BiOX can effectively split and transit the photogenerated electrons and holes [61–63]. However, BiOX as a photocatalyst could be employed for the degradation of organic pollutants alone as its positive CB (conduction band) potential restricts it from being used for other photocatalytic applications such as H$_2$ generation, CO$_2$ reduction, N$_2$ fixation and organic synthesis.

On the other hand, non-stoichiometric Bi$_x$O$_y$X$_z$ with increased Bi content are reported to promote the reduction power of photogenerated electrons and increase the thermodynamic force for initiating many reduction reactions that were impossible to be carried out using BiOX [64,65]. For instance, compared with BiOX, the changes in the Bi, O, and X proportions result in the variation of orbital hybridization and uplifting of the bottom of the CB, leading to the water splitting for H$_2$ generation as was reported in Bi$_4$O$_{5\,\times\,2}$ (X = Br and I) [66,67]. In addition to H$_2$ generation, the increased CB also promoted photocatalytic molecular oxygen activation in Bi$_{24}$O$_{31}$Cl$_{10}$. Further, the Bi-rich Bi$_x$O$_y$X$_z$ possesses enhanced light-harvesting ability that is attributed to the modulated band structure, thus breaking the bottleneck of limited photoabsorption caused from the wide bandgap energy of BiOCl and BiOBr [68]. The higher photon absorption efficiency of Bi$_x$O$_y$X$_z$ in comparison to BiOX induces greater electric field intensity, which in turn leads to large dipole moment. The larger dipole moment and wider interlayer spacing in Bi$_x$O$_y$X$_z$ boosted by the large

polarization force and polarization space lead to increased IEF, which in turn enhances the separation efficiency of the photogenerated charge carriers.

Electronic band structure, redox levels of the CB and VB and the bandgap energy of g-C$_3$N$_4$ were studied both theoretically and experimentally. Theoretical calculations estimated the bandgap energy of the melem molecule, polymeric melon and fully condensed g-C$_3$N$_4$ to be 3.5, 2.6 and 2.1 eV, respectively [69–71]. The bandgap energy value of 2.6 eV calculated for polymeric melon was consistent with the experimentally measured value of 2.7 eV for defect containing bulk g-C$_3$N$_4$ [70]. The CBM and VBM positions for g-C$_3$N$_4$ estimated through the density functional theory were −1.12 and +1.57 eV, respectively. Interestingly, the experimental investigations through the valence band X-ray photoelectron spectroscopy confirmed the VBM position of g-C$_3$N$_4$ at +1.53 eV, which was almost consistent with the theoretical calculations [72]. Therefore, the position of the CBM (−1.12 eV) is predicted to be satisfactory for H$_2$ generation, while that of the VBM provides a thermodynamic driving force for O$_2$ evolution reaction. Wavefunction studies revealed that the VB and CB of g-C$_3$N$_4$ serving as independent sites for the oxidation and reduction reactions during water splitting are mainly driven by the nitrogen P$_z$ orbitals and carbon P$_z$ orbitals, respectively. Further, the redox potential levels of water calculated by ab initio thermodynamics indicated that both the reduction and oxidation level of water splitting are located within the bandgap of g-C$_3$N$_4$ [69]. Another theory using the many-body Green's function reported that lone pair electrons of nitrogen atoms are mainly responsible for the formation of the VB and electronic structure [20,73]. Additionally, it was proposed that the N 2p orbital overlapping the C 2p orbital mainly contributes to the VB and CB of g-C$_3$N$_4$, respectively [74]. As observed from Figure 3, the CB position of g-C$_3$N$_4$ and many of the Bi$_x$O$_y$X$_z$ are suitable for photocatalytic H$_2$ generation, CO$_2$ reduction, N$_2$ fixation and molecular oxygen activation in addition to their potential to be utilized in the degradation of organic pollutants. Further, it is evident from Figure 3 that the VB and CB levels of g-C$_3$N$_4$ match well with those of BiOX and Bi$_x$O$_y$X$_z$ for the fabrication of efficient 2D/2D heterojunction photocatalysts.

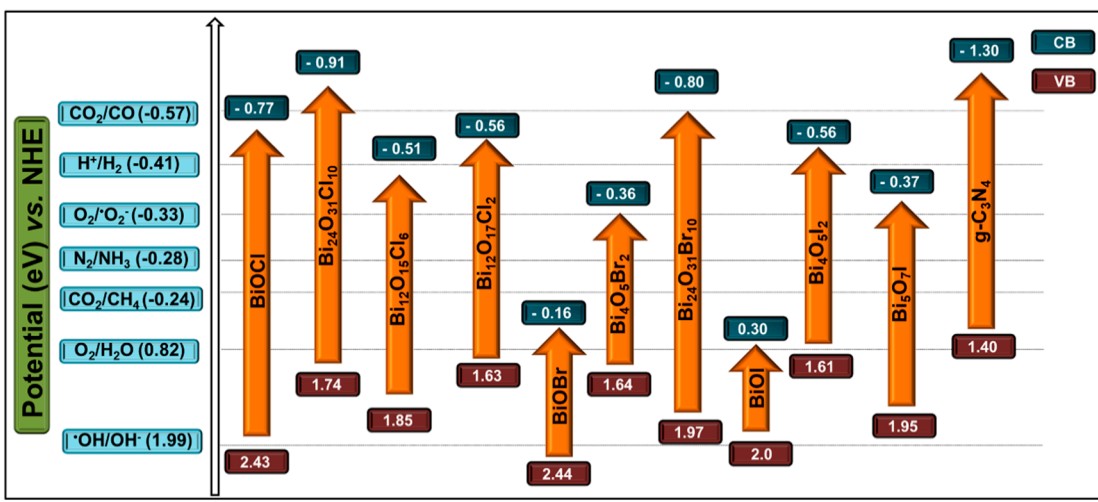

**Figure 3.** Conduction and valence band (CB and VB) positions of g-C$_3$N$_4$, BiOX, and some of the Bi$_x$O$_y$X$_z$ photocatalysts *vs.* Normal Hydrogen Electrode (NHE) at pH = 7. The redox potentials of different chemical reactions are compared in this figure.

## 3. Fabrication of BiOX/Bi$_x$O$_y$X$_z$-g-C$_3$N$_4$ Heterojunction Photocatalysts

Fabrication is a significant step involved in tailoring the band structure of photocatalysts due to its dependence on the chemical composition. Morphology, shape, size and surface area are some of the critical parameters that play a determinant role in the adsorption properties and photocatalytic activity. Benefiting from the large specific surface area, 2D semiconductors can provide abundant surface active sites. More importantly, the greatly reduced thickness of 2D semiconductors relative to bulk counterparts shortens the

bulk carrier diffusion distance and improves the charge separation. Further, the surface charge separation efficiency is enhanced by the creation of surface defects such as oxygen vacancies during the fabrication of the 2D semiconductors.

The typical bismuth metal precursors utilized for synthesizing BiOX and $Bi_xO_yX_z$ are $Bi(NO_3)_3 \cdot 5H_2O$, $NaBiO_3 \cdot 2H_2O$, $Bi_2O_3$, $BiCl_3$ and $BiI_3$, while the halogen precursors include KX, NaX, HX, CTAX (X = Cl, Br or I) and ionic liquids containing halogen elements. Various solution based fabrication techniques such as the electrostatic self-assembly approach, the hydrothermal method, the ionic liquid-assisted method, the impregnation method, the solid-phase calcination step, the solvothermal method, precipitation, the reflux process and the ultrasound-assisted water bath technique are used in the synthesis of BiOX and $Bi_xO_yX_z$. On the other hand, the precursors used for synthesizing $g-C_3N_4$ through the most typical thermal polycondensation method are urea, thiourea, melamine and dicyandiamide. The fabrication of $g-C_3N_4$-BiOX/$Bi_xO_yX_z$ as 2D-2D heterojunction photocatalysts is usually achieved by growing BiOX/$Bi_xO_yX_z$ on the surface of pre-synthesized $g-C_3N_4$. An overview of the various synthetic methods and the corresponding growth mechanism is presented in detail.

### 3.1. In Situ Self-Assembly

The effective use of electrostatic forces in the self-assembly and fabrication of nanostructures is gaining significance owing to their flexibility to work at room temperature and also due to their ability to offer rigid interface among the integrated components. For example, Yang et al. synthesized BiOBr/$g-C_3N_4$ composite through the in situ self-assembly process based on electrostatic interaction between the precursors followed by their precipitation. In a typical process, pre-synthesized $g-C_3N_4$ was protonated by treating it with HCl solution for converting its surface charge from negative to positive. The protonated $g-C_3N_4$ was then added to KBr solution such that the $Br^-$ gets attracted to its surface and subsequently undergoes a precipitation reaction to form BiOBr with the addition of $Bi(NO_3)_3 \bullet 5H_2O$ solution [62]. Therefore, the BiOBr layer was favourably formed on the positively charged surface of $g-C_3N_4$ and led to the formation of a tightly bound 2D-2D semiconductor heterojunction. Similarly, a p-n heterojunction between flower-like BiOI sheets and $g-C_3N_4$ nanoparticles was constructed through an electrostatic self-assembly of $g-C_3N_4$ nanoparticles, wherein the zeta potential of BiOI sheets was −11.1 mV and that of the $g-C_3N_4$ nanoparticles was +21.5 mV [75]. The measured values of zeta potential clearly indicated that the heterojunction formed between them was via the electrostatic self-assembly process.

### 3.2. Hydrothermal and Solvothermal Synthesis

Hydrothermal synthesis refers to process of heating water above its boiling point in a sealed reaction vessel to create supercritical fluid that in turn facilitates the precipitation or crystallization of inorganic materials under auto-generated pressure. The hydrothermal synthesis of nanostructured materials is similar to the processes governing the formation of minerals under the earth's crust that have been experimentally studied by geologists. The hydrothermal process can be used for dissolving and recrystallizing a substance that is poorly soluble or insoluble under normal conditions. Typically, an aqueous mixture of precursors sealed in a stainless steel autoclave heated above the boiling point of water results in the single-step production of highly crystalline materials due to the synergistic effect of high temperature and pressure [76]. The merits of hydrothermal synthesis are the enhanced crystallinity of synthesized materials without the need for further calcination, and easy control of the morphology and phase composition by controlling the temperature and reaction time. Under hydrothermal conditions, reactants enter the solution in the form of ions and are adsorbed, decomposed and desorbed at the growth interface before crystallizing. Solvothermal synthesis is analogous to the hydrothermal synthesis process, except for the fact that water is replaced by an organic solvent such as ethanol, ethylene glycol, etc. Adjusting the thermodynamic and kinetic parameters of the solvothermal

synthesis reaction such as the concentration of the reactant precursors, reaction time, pH and temperature aids in controlling the size, shape, uniformity, dimensionality, phase and facets of the inorganic materials [77]. Therefore, the hydrothermal and solvothermal reactions can possibly ensure the intimate interface contact between $BiOX/Bi_xO_yX_z$ and $g-C_3N_4$ for promoting the rapid transport of photogenerated charge carriers across the interface.

Xiao et al. reported the synthesis of thirteen kinds of $BiOX$ and $Bi_xO_yX_z$, viz., $BiOI$, $Bi_4O_5I_2$, $Bi_7O_9I_3$, $Bi_5O_7I$, $BiOBr$, $Bi_4O_5Br_2$, $Bi_{24}O_{31}Br_{10}$, $Bi_3O_4Br$, $BiOCl$, $Bi_{12}O_{15}Cl_6$, $Bi_{24}O_{31}Cl_{10}$, $Bi_3O_4Cl$, and $Bi_{12}O_{17}Cl_2$ through a general one-pot hydrothermal route by reacting different compositions of $Bi_2O_3$ and $KX$ (X = Cl, Br and I) with nitric acid, and it was the first of its kind [78]. Since then, hydrothermal synthesis for the fabrication of $BiOX$ and $Bi_xO_yX_z$ with various morphologies such as microspheres, microflowers, and microdisks (3D hierarchical structures) was achieved and comprised of three main growth steps: (i) the creation of $BiOX$ nuclei, (ii) the growth of 2D nanosheets through the dissolution-renucleation process, and (iii) the formation of 3D nanostructures from the oriented attachment of 2D nanosheets under the influence of an electrostatic multipole field [79,80]. The hydrothermal method with L-lysine as a bio-template was employed in the fabrication of $BiOBr/g-C_3N_4$ semiconductor heterojunction. Flake-like $g-C_3N_4$ was pre-synthesized by the thermal polycondensation of melamine followed by sonochemical treatment in $NH_4Cl$ solution and subsequent sintering at 550 °C. $BiOBr$ microspheres with various mass ratios (5, 10, 15, 20 and 25%) were grown in situ on flake-like $g-C_3N_4$ under hydrothermal conditions with $Bi(NO_3)_3\bullet5H_2O$, $NaBr$ as precursors and L-lysine as the bio-template. Experimental investigation using TEM revealed that $BiOBr$ microspheres synthesized with L-lysine as the template exhibited a loose structure with a larger percentage of exposed nanosheets that enhanced the amount of active sites for the degradation of organic pollutants in comparison to those synthesized without L-lysine [81]. Similarly, the hierarchical nanostructures of $Bi_xO_yX_z$ synthesized hydro/solvothermally with interconnected porous networks were reported to accelerate molecular diffusion/transport, enhance the overall light utilization efficiency, possess a large accessible surface area and provide better permeability, which could not only furnish adequate active adsorption sites and photocatalytic reaction sites, but also contributed to uniformly distributing the active sites in the fabricated photocatalysts [82]. The solvothermal method employed for synthesizing $Bi_5O_7Br$ nanotubes using oleylamine as the solvent exhibited good visible light absorption and created oxygen vacancies on the surface that were beneficial for the stable photoreduction process [83]. Liu et al. reported the solvothermal synthesis of a 3D hierarchical structure of $g-C_3N_4@Bi/BiOBr$ with ternary heterojunction employing ethylene glycol as the solvent and reducing agent, which exhibited notably high photocatalytic activity for degrading organic pollutants [84]. Similarly, ethylene glycol assisted solvothermal synthesis reported by Ji et al. for the fabrication of ultrathin $Bi_4O_5Br_2$ nanosheets dispersed over layered $g-C_3N_4$ also exhibited higher photocatalytic activity for ciprofloxacin decomposition under visible light irradiation [85]. Another report on solvothermal synthesis was reported for the synthesis of $g-C_3N_4/I^{3-}-BiOI$ heterojunction semiconductor using self-stabilized $I_3^-/I^-$ as a redox mediator that efficiently strengthened the interaction between porous $g-C_3N_4$ and ultrathin $BiOI$, thereby enhancing their photocatalytic activity in $CH_3SH$ oxidation [63].

### 3.3. Ionic Liquid-Assited Method

Solvent plays a prominent role in controlling the morphology of the nanostructured materials synthesized through the liquid phase synthesis techniques. Though organic solvents employed in various synthetic techniques are immensely useful in the shape and size controlled synthesis of nanostructured materials, some of their drawbacks such as poor solubility of inorganic precursors, low boiling point, high vapor pressure, high toxicity and flammable/explosive nature make them unpopular. Therefore, ionic liquids are gaining significant attention as a green medium for the synthesis of inorganic materials

due to the growing environmental awareness. Low melting point, high chemical and thermal stability, high polarity for solubilizing a wide range of compounds, and ability to act as an ionic halide source are the attractive properties of ionic liquids. Various semiconductor photocatalysts have been synthesized using ionic liquid as solvent, and since they possess halogens in their functional groups, they are more suited to the preparation of $BiOX/Bi_xO_yX_z$ [86]. For example, Xia et al. reported the synthesis of ultrathin $g-C_3N_4/Bi_4O_5I_2$ layered nanojunctions using [Hmim]I (1-hexyl-3-methylimidazolium iodide) ionic liquid. Highly reactive ionic liquid acted as the iodine source, also served as the capping agent for the formation of ultrasmall $Bi_4O_5I_2$ nanosheets and facilitated the wide distribution over ultrathin $g-C_3N_4$. The growth of ultrasmall $Bi_4O_5I_2$ and their wide distribution over the ultrathin $g-C_3N_4$ promoted the construction of a tight heterojunction under hydrothermal conditions [87].

Similarly, $g-C_3N_4/BiOBr$ microspheres were synthesized by the dispersion of $g-C_3N_4$ to a solution made by dissolving $Bi(NO_3)_3 \bullet 5H_2O$ in ethanol containing a stoichiometric amount of ionic liquid $[C_{16}mim]Br$ (1-hexadecyl-3-methylimidazolium bromide). During the reaction, the ionic liquid $[C_{16}mim]Br$ acted as the solvent, reactant, template and most importantly as a dispersing agent, which ensured the better dispersion of $g-C_3N_4$ in the aqueous solution due to electrostatic attraction. As observed from Figure 4, the FESEM and TEM micrographs of the solvothermally synthesized $g-C_3N_4/BiOBr$ composites exhibited relatively uniform 3D flower-like microspheres with self-assembled nanosheets on their surface, indicating the wide distribution of $g-C_3N_4$ on the surface of BiOBr [88,89].

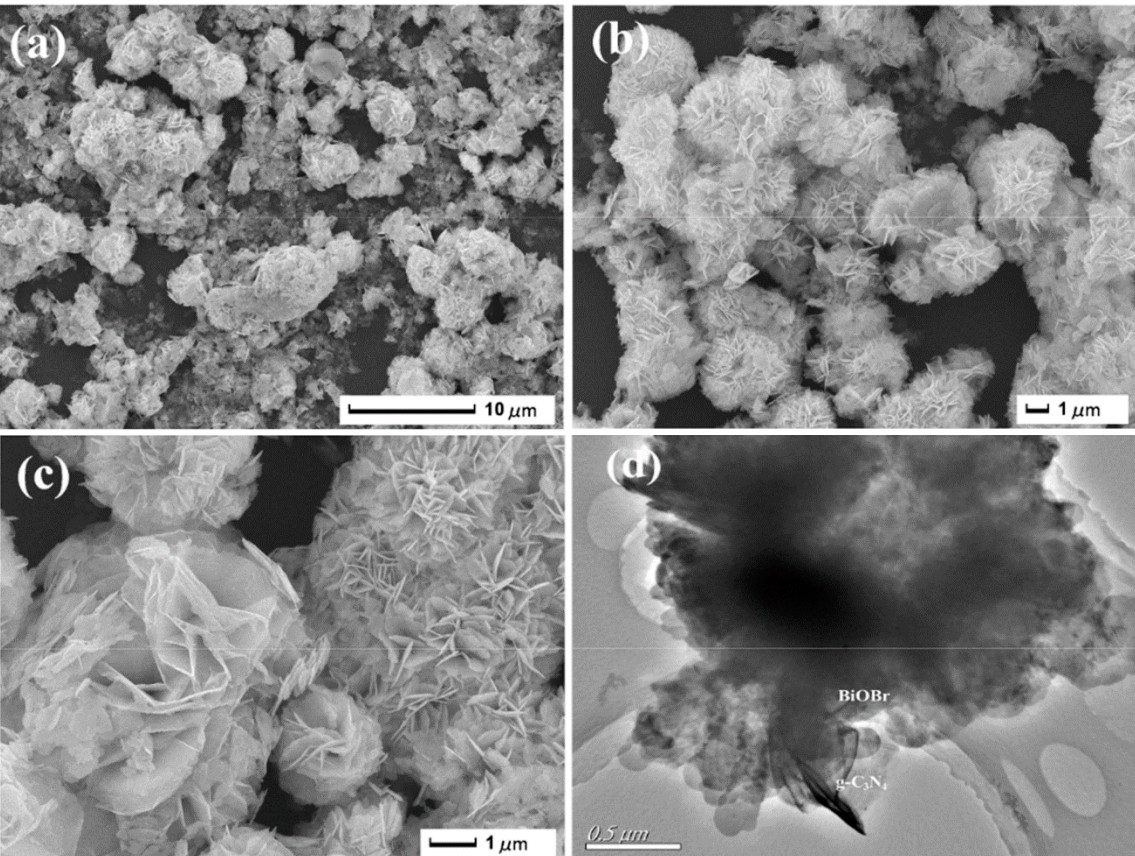

**Figure 4.** (**a**) Low magnification and (**b**,**c**) magnified FESEM and (**d**) TEM micrographs of $g-C_3N_4/BiOBr$ microspherical composites synthesized by ionic liquid assisted solvothermal method. Reprinted from Ref. [88] with permission from The Royal Society of Chemistry.

### 3.4. Precipitation Technique

Precipitation is a simple, cost-effective and rapid process of synthesizing semiconductor photocatalysts that can be easily replicated on a larger scale for industrial applications. Further, it is an eco-friendly route that hardly requires any hazardous organic solvents and treatments under high pressure or temperature [90]. The precipitation synthesis of BiOX typically involves the dropwise addition of halide (KX or NaX, where X = Cl, Br and I) solution into a solution of bismuth salt ($BiCl_3$, $Bi(NO_3)_3$ or $Bi_2O_3$) under acidic conditions. For instance, Ren et al. reported the preparation of three series of $BiOM_xR_{1-x}$ (M, R = Cl, Br, I) solid solutions with 3D nanostructured morphology and adjustable bandgap energy through a low-temperature precipitation technique [59]. Appropriately, adjusting the amount of solute and the solvent in the solid solutions led to the formation of $BiOM_xR_{1-x}$ photocatalysts that could absorb visible light in the range 359–675 nm with a bandgap energy ranging from 3.3–1.7 eV. Composite heterojunctions of $g\text{-}C_3N_4/Bi_{12}O_{17}Cl_2$ were prepared by dispersing pre-synthesized $g\text{-}C_3N_4$ into an ethanol solution of $BiCl_3$ at pH 2. The dropwise addition of freshly prepared aqueous NaOH solution into the ethanol solution containing the mixture led to the formation of $g\text{-}C_3N_4/Bi_{12}O_{17}Cl_2$ while the pH reached 14 [60]. Chen et al. reported the synthesis of hierarchical hexagonal plates of $Bi_{24}O_{31}Br_{10}$ through co-precipitation and subsequent solvothermal treatment in ethylene glycol, which produced a hierarchical structure by the process of dissolution-recrystallization of 1D $Bi_{24}O_{31}Br_{10}$ nanobelts [61]. In another study, a $BiOI\text{-}BiOCl/g\text{-}C_3N_4$ ternary composite was synthesized by a template-free precipitation method using $NH_3$ solution as the precipitating agent, wherein thin layers of $g\text{-}C_3N_4$ acted as a bed for anchoring BiOI and BiOCl nanosheets for the formation of an efficient heterojunction semiconductor [91].

### 3.5. Reflux Process

Reflux based synthesis is based on the thermal energy supplied for the progress of the reaction over long periods of time. The phase and morphology of the synthesized nanostructured materials are directly dependent on parameters, viz., the order in which the precursors are added, reflux time and cooling rate [47]. Mousavi et al. employed the reflux technique for the fabrication of $g\text{-}C_3N_4/Fe_3O_4/BiOI$ nanocomposites. As the first step, $Fe_3O_4$ nanoparticles were deposited on the surface of pre-synthesized $g\text{-}C_3N_4$ to form $g\text{-}C_3N_4/Fe_3O_4$. Next, BiOI was synthesized over the surface of $g\text{-}C_3N_4/Fe_3O_4$ by a precipitation reaction between $Bi(NO_3)_3 \bullet 5H_2O$ and NaI, followed by refluxing for 30 min at 96 °C [92]. Similarly, the fabrication of $g\text{-}C_3N_4/carbon\ dots/BiOCl$ and $g\text{-}C_3N_4/carbon\ dots/BiOBr$ heterojunction photocatalysts was also reported by employing the reflux process [93,94].

### 3.6. Solid-State Calcination

Solid-state calcination is a viable method for the preparation of materials without the utilization of water. Weak van der Waals interaction existing between halogen atoms results in the phase transition from BiOX to $Bi_xO_yX_z$ during the process of calcination due to the removal of unstable halogen. Therefore, $Bi_xO_yX_z$ materials are prepared by the high temperature treatment of the precursors mixed with appropriate stoichiometric ratio. For example, Di et al. reported the preparation of $Bi_{12}O_{17}Cl_2$ by calcining a mixture of $Bi_2O_3$ and BiOCl in stoichiometric proportions at 650 °C for 10 h [65]. A similar process was reported for synthesizing $Bi_3O_4Br$ by the calcination of $Bi_2O_3$ and BiOBr mixture at 650 °C for 10 h [95]. Additionally, the solid-state calcination method was employed in the fabrication of $Bi_3O_4Cl/g\text{-}C_3N_4$ heterojunction that was reported to have a tight face-to-face connection between the semiconductors for improved photocatalytic activity [96].

### 3.7. Sonochemical Synthesis

Sonochemical (also known as ultrasound-assisted) synthesis is a versatile approach that utilizes the high intensity ultrasound for the production of nanostructured inorganic materials in a controllable fashion, which are often unattainable through the conventional

methods [97]. Water and ionic liquids are typically used as replacements for volatile and toxic organic solvents [98]. In comparison to the chemical reactions progressing through the supply of common energy sources (such as heat, light, electric potential, radiation, etc.), the ultrasonic irradiation provides an unusual reaction condition that leads to acoustic cavitation (i.e., the formation, growth and implosive collapse of bubbles in liquids), which drives the rapid nucleation and growth of the inorganic materials. For example, Liu et al. reported the synthesis of a g-$C_3N_4$/BiOBr heterojunction photocatalyst through the sonochemical synthesis technique [99]. A solution of $Bi(NO_3)_3 \bullet 5H_2O$ dissolved in ethylene glycol was mixed with DI water containing pre-synthesized g-$C_3N_4$ under ultrasound irradiation at 40 °C for 2 h to form a uniform suspension, to which a stoichiometric proportion of NaBr and PVP were added dropwise and heated to 80 °C for 3 h. TEM micrographs of the pristine g-$C_3N_4$, pristine BiOBr and sonochemically synthesized g-$C_3N_4$/BiOBr are shown in Figure 5. A schematic representation of the 2D-2D heterojunction (Figure 5g) and the elemental maps (Figure 5h) confirming the deposition of BiOBr over g-$C_3N_4$ is also shown in Figure 5.

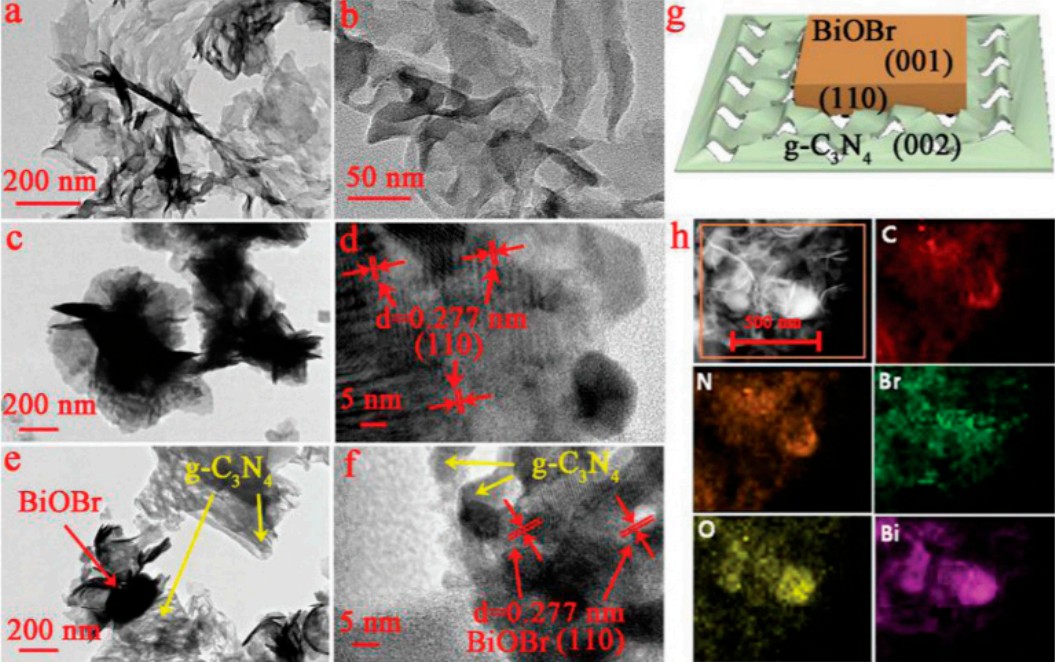

**Figure 5.** TEM micrographs of (**a**,**b**) pristine g-$C_3N_4$, (**c**,**d**) pristine BiOBr and (**e**,**f**) g-$C_3N_4$/BiOBr; (**g**) schematic representation of growth of BiOBr layer over the surface of g-$C_3N_4$ forming the heterojunction photocatalyst and (**h**) elemental maps confirming the growth of BiOBr over g-$C_3N_4$. Reprinted from Ref. [99] with permission from The Royal Society of Chemistry.

## 4. Photocatalytic Activity

### 4.1. Photocatalytic Degradation of Organic and Inorganic Contaminats

BiOX/$Bi_xO_yX_z$ photocatalysts have demonstrated admirable performance in the degradation of various organic and inorganic contaminants such as methyl orange, rhodamine B, methylene blue, acid orange, microcystin-LR, 2,4 dichlorophenol, bisphenol-A, tetracycline hydrochloride, phenol, carbamazepine, levofloxacin, metronidazole, fuchsine, methyl mercapton, sulfamethoxazole, mercury, chromium, etc. In general, the photocatalytic reaction for the degradation of organic contaminants involves three simultaneous steps, viz., photoexcitation for the generation of charge carriers ($e^-$ and $h^+$) at the CB and VB, the separation of charges and their transfer to the active sites on the semiconductor surface, the formation of radical species by the ionization of water, i.e., reaction of the holes ($h^+$) with hydroxyl ions ($OH^-$) to produce hydroxyl radicals ($^{\bullet}OH$) and reaction of the electrons ($e^-$) with the superoxide anion radicals ($^{\bullet}O_2^-$), which subsequently react with

the organic contaminants adsorbed on the photocatalyst surface [100,101]. Di et al. synthesized ultrathin $Bi_4O_5Br_2$ and BiOBr nanosheets and studied their capability to degrade ciprofloxacin under visible light. Lower energy bandgap (2.33 eV) and a more negative CB position of ultrathin $Bi_4O_5Br_2$ nanosheets facilitated the improved electronic transition, the generation of extra charge carriers and the formation of more $\bullet O_2^-$ radicals that collectively enabled it to display a maximum rate constant of 0.0113 $min^{-1}$, which was 1.9 times higher than ultrathin BiOBr nanosheets [102]. Wang et al. synthesized $Bi_{24}O_{31}Br_{10}$ nanosheets with thicknesses of 40, 85 and 130 nm through the solvothermal method and utilized them for the photodegradation of tetracycline hydrochloride under visible light irradiation. The three $Bi_{24}O_{31}Br_{10}$ nanosheets with 40 nm thickness demonstrated 95% degradation of tetracycline hydrochloride within 90 min, in comparison to the thicker counterparts. The enhanced photocatalytic activity of $Bi_{24}O_{31}Br_{10}$ nanosheets with 40 nm thickness was attributed to lattice defects formed by bromine vacancies that subsequently improved the charge carrier density, charge separation and transportation [103]. A BiOBr-g-$C_3N_4$ heterojunction photocatalyst synthesized through a single-step chemical bath method exhibited enhanced photodegradation of 10 ppm rhodamine B under visible light in comparison to pristine g-$C_3N_4$, pristine BiOBr and a composite formed by mixing g-$C_3N_4$ and BiOBr in 1:1 weight ratio. The enhanced performance of the BiOBr-g-$C_3N_4$ photocatalyst was attributed to the perfect coupling between the BiOBr-{001} and g-$C_3N_4$-{002} facets, which facilitated the unhindered transport of the photogenerated charges while curbing their recombination [104]. Sphere-like g-$C_3N_4$/BiOI composite photocatalysts synthesized using ionic liquids exhibited excellent photocatalytic activity in the degradation of rhodamine B, methylene blue, methyl orange, bisphenol A and 4-chlorophenol under visible light irradiation. Among the various composite photocatalysts, the 15 wt% g-$C_3N_4$/BiOI exhibited optimal performance in comparison to pristine BiOI, which was attributed to the heterojunction formed between g-$C_3N_4$ and BiOI that effectively separated the photogenerated charge carriers and enhanced the interfacial charge transfer as evidenced through its photocurrent response [105]. Liu et al. reported the fabrication of g-$C_3N_4$/$Bi_5O_7I$ composite photocatalysts by the thermolysis of melamine with pre-synthesized BiOI at 520 °C for 4 h [106]. Interestingly, during thermolysis BiOI was transformed to $Bi_5O_7I$ and a strong interfacial contact was established with g-$C_3N_4$ due to in situ co-crystallization, which enabled it to exhibit excellent performance in the photodegradation of rhodamine B and phenol under visible light irradiation due to faster charge migration and separation over the heterojunction. The results revealed that $h^+$ and $\bullet O_2^-$ were the primary active species, and the rate of photodegradation of rhodamine B using 30 wt% g-$C_3N_4$/$Bi_5O_7I$ at 1.12 $h^{-1}$ was ~15 and 3 times higher than that of pristine g-$C_3N_4$ and $Bi_5O_7I$, respectively. In another study, microspheres of g-$C_3N_4$/$Bi_5O_7I$ synthesized through the hydrothermal method using ethylene glycol as the solvent exhibited enhanced photodegradation of methyl orange and rhodamine B with rate constants 0.084 $min^{-1}$ and 0.197 $min^{-1}$, respectively. The results of scavenger studies and electron spin resonance spectroscopy confirmed that $\bullet O_2^-$ was the primary active species, which could only have been generated if the transfer mechanism was based on the Z-scheme heterojunction [107]. The visible light photocatalytic oxidation of hazardous gas-phase mercury ($Hg^0$) to divalent mercury ($Hg^{2+}$) for its easy removal was reported using g-$C_3N_4$/$Bi_5O_7I$ nanosheets doped with $Yb^{3+}$ [108]. As observed from Figure 6a, the mercury removal efficiency of g-$C_3N_4$/$Bi_5O_7I$ doped with $Yb^{3+}$ was 79.01% and 42.02%, respectively, under visible and near infrared light radiation, while the efficiency under near infrared light was just 13.3% without $Yb^{3+}$ doping. Scavenger studies and electron spin resonance spectroscopy revealed that $\bullet O_2^-$ and $\bullet OH$ were the primary active species responsible for the oxidation of gas phase $Hg^0$, while the mechanism of charge transfer was based on the Z-scheme heterojunction with enhanced separation of electrons due to the formation of a new energy band below the CB of $Bi_5O_7I$ as a result of doping $Yb^{3+}$, as depicted in Figure 6b.

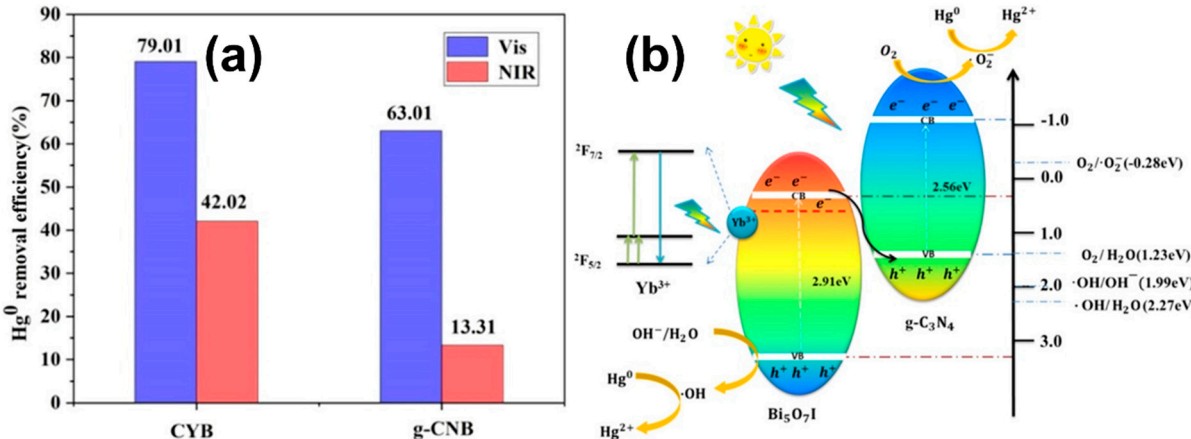

**Figure 6.** (**a**) Mercury removal efficiency under visible and near infrared light excitation in the presence of g-$C_3N_4$/$Yb^{3+}$-$Bi_5O_7I$ (CYB) and g-$C_3N_4$-$Bi_5O_7I$ (g-CNB) as photocatalysts, (**b**) schematic depicting the mechanism of charge transfer in the Z-scheme heterojunction g-$C_3N_4$/$Yb^{3+}$-$Bi_5O_7I$ during the photocatalytic oxidation of $Hg^0$. Reprinted from Ref. [108] with permission from American Chemical Society.

Zhang et al. reported the fabrication of a heterojunction photocatalyst by the in situ hydrothermal growth of $Bi_7O_9I_3$ on ultrathin g-$C_3N_4$ for the degradation of doxycycline hydrochloride under visible light. Microspheres of $Bi_7O_9I_3$/g-$C_3N_4$ exhibited a photodegradation efficiency of ~80% that was ~2 and 5.4 times greater than pristine $Bi_7O_9I_3$ and g-$C_3N_4$, respectively, which could be attributed to their large surface area (68.55 $m^2$ $g^{-1}$) and enhanced charge generation/separation in the heterojunction. Scavenger studies and electron spin resonance spectroscopy revealed that $^\bullet O_2^-$ and $^\bullet OH$ were the primary active species that were predominantly involved in breaking the stable structure of doxycycline hydrochloride, while all the experimental data and characterization evidence confirmed that the mechanism of photodegradation followed direct Z-scheme heterojunction [109]. In another study, a g-$C_3N_4$ modified $Bi_4O_5I_2$ composite prepared in situ by the thermal treatment of a g-$C_3N_4$/$Bi_4O_5I_2$ precursor at 400 °C for 3 h exhibited enhanced photocatalytic performance in the degradation of methyl orange under visible light with a degradation rate of 0.164 $min^{-1}$, which was 3.2 and 82 times enhanced in comparison to pristine $Bi_4O_5I_2$ and g-$C_3N_4$, respectively [110]. A summary of the typical synthesis methods and photocatalytic performance of $BiOX$/$Bi_xO_yX_z$-g-$C_3N_4$ heterojunction photocatalysts involved in the degradation of various organic pollutants is presented in Table 1. Further, the details corresponding to the mechanism of photogenerated charge transfer during the degradation of organic pollutants are briefly explained in Section 5.5.

**Table 1.** Summary of the degradation of organic contaminants in the presence of $BiOX$/$Bi_xO_yX_z$-g-$C_3N_4$ heterojunction photocatalysts reported in the literature.

| Synthesis Method | Precursors | Morphology | Contaminant Parameters | Light Source | Heterojunction Type | Significance of the Result | Ref. |
|---|---|---|---|---|---|---|---|
| | | | $BiOX$-g-$C_3N_4$ | | | | |
| | | | $BiOI$-g-$C_3N_4$ | | | | |
| Solid-phase calcination | $Bi(NO_3)_3 \cdot 5H_2O$, KI and $C_3H_6N_6$ | Layers of g-$C_3N_4$ grown on the surface of BiOI microspheres | Microcystin-LR (5 ppm) | 350 W Xe lamp ($\lambda > 420$ nm) | Direct solid-state Z-scheme | Optimized content of g-$C_3N_4$ over BiOI for high activity was found to be 4 wt% | [111] |
| Electrostatic self-assembly | $Bi(NO_3)_3 \cdot 5H_2O$, KI, $C_3H_6N_6$, $H_2SO_4$, $HNO_3$, and $C_2H_6O_2$ | g-$C_3N_4$ nanoparticles on flower-like BiOI nanosheets | Methyl orange (10 ppm) | 300 W Xe lamp ($\lambda > 420$ nm) | p-n | Surficial dispersive heterojunctions were beneficial for degradation of MO | [75] |

**Table 1.** *Cont.*

| Synthesis Method | Precursors | Morphology | Contaminant Parameters | Light Source | Heterojunction Type | Significance of the Result | Ref. |
|---|---|---|---|---|---|---|---|
| Simple precipitation | $Bi(NO_3)_3 \cdot 5H_2O$, KI, $C_3H_6N_6$, $C_2H_6O_2$ and CTAB | Thin nanosheets of BiOI lie on the surface of g-$C_3N_4$ | 2,4-dichlorophenol (10 ppm) Bisphenol A (10 ppm) Rhodamine B (100 ppm) Tetracycline hydrochloride (10 ppm) | 300 W Xe lamp (λ > 420 nm) | p-n | Top-top facets of BiOI (001)/g-$C_3N_4$ (002) promoted generation of $^1O_2$ and $^\bullet O_2^-$ accounting for excellent photocatalytic activity | [112] |
| In situ precursor transformation | $CO(NH_2)_2$, $Bi(NO_3)_3 \cdot 5H_2O$, KI and $C_2H_6O_2$ | Numerous quantum-sized nanoparticles are uniformly dispersed across the g-$C_3N_4$ nanosheets | Phenol (100 ppm) | 60 W LED lamp (λ > 420 nm) | Direct Z-scheme | Increase in the electron density on BiOI led to internal electric field formation favouring Z-scheme configuration | [113] |
| Solvothermal | $Bi(NO_3)_3 \cdot 5H_2O$, KI, $C_3H_6N_6$ and $C_2H_6O_2$ | BiOI nanoplates are irregularly dispersed over the surface of g-$C_3N_4$ nanosheets | Rhodamine B (20 ppm) | 300 W Xe lamp (λ > 420 nm) | p-n | Charge transfer mode in the BiOI/g-$C_3N_4$ followed the double-transfer mechanism | [114] |
| Solvothermal | $Bi(NO_3)_3 \cdot 5H_2O$, KI, $HNO_3$, $C_3H_6N_6$ and $CH_4N_2S$ | Thin nanosheets of BiOI composited with wrinkled nanosheets of g-$C_3N_4$ | Methylene blue (20 ppm) | 50 W, 410 nm LED light arrays | Direct Z-scheme | Strong IEF at interface occurred due to difference in their Fermi energies was proved by DFT calculations | [115] |
| Ultrasonication-assisted | $C_3H_6N_6$, $Bi(NO_3)_3 \cdot 5H_2O$, KI, $C_2H_5OH$ | BiOI particles are grown over the surface of g-$C_3N_4$ sheets | Cr (VI) (10 ppm) | 500 W Xe lamp | Z-scheme | The non-radiative recombination process of photoinduced carriers at the interface was confirmed by photoluminescence and ESR | [116] |
| BiOCl-g-$C_3N_4$ | | | | | | | |
| Simple calcination | $Bi(NO_3)_3 \cdot 5H_2O$, $CH_4N_2O$, $C_6H_{14}O_6$ and NaCl | g-$C_3N_4$ nanosheets acted as substrate for compactly anchoring BiOCl nanoplates | Methyl orange (10 ppm) | 300 W Xe lamp | Binary heterojunction | Large contact surface of 2D hybrid structure was efficient in solving detrimental photoinduced carrier recombination | [117] |
| Solvothermal | $Bi(NO_3)_3 \cdot 5H_2O$, NaCl, PVP, K-30, $CH_4N_2O$ and $C_3H_8O_3$ | Ultrathin nanosheets of BiOCl are covered by 2D g-$C_3N_4$ layers stacked in the form of multi-slice structure | 4-chlorophenol (10 ppm) | 300 W short-arc Xe lamp | Binary heterojunction | Introduction of oxygen vacancies brings a new defect level for increased photoabsorption | [118] |
| Hydrothermal | $C_3H_6N_6$, $NH_4Cl$, $Bi(NO_3)_3 \cdot 5H_2O$ and KCl | Smooth surface of BiOCl nanodiscs turned rough after loading ultrathin g-$C_3N_4$ nanosheets | Rhodamine B (10 ppm) | 300 W Xe lamp | p-n | Photosensitization of RhB played critical role in degradation process over BiOCl under visible light | [119] |

**Table 1.** *Cont.*

| Synthesis Method | Precursors | Morphology | Contaminant Parameters | Light Source | Heterojunction Type | Significance of the Result | Ref. |
|---|---|---|---|---|---|---|---|
| Ionic liquid-assisted | $C_2H_4N_4$, $C_2H_6O_2$, $Bi(NO_3)_3·5H_2O$ and $[C_{16}mim]Cl$ | Spherical microstructures with large number of smaller nanosheets of BiOCl and g-$C_3N_4$ | Rhodamine B (10 ppm) | 300 W Xe lamp | p-n | $[C_{16}mim]Cl$ having positive polarity improved the dispersity of g-$C_3N_4$ | [89] |
| In situ surfactant-free | $C_3H_6N_6$, $Bi(NO_3)_3·5H_2O$, HCl and $C_2H_5OH$ | Irregular elliptical BiOCl nanosheets are grown over the surface of g-$C_3N_4$ sheets | Rhodamine B (10 ppm) | 300 W Xe lamp | Binary heterojunction | The appropriate proportion of BiOCl in heterojunction and large surface area with higher adsorption capacity provided larger photoactive sites for photodegradation of RhB | [120] |
| Microwave-assited | $Bi(NO_3)_3·5H_2O$, KCl, $C_2H_6O_2$, $C_2H_4N_4$ | Microspheres assembled by nanosheets | Carbamazepine (2.5 ppm) | LED lamp ($\lambda >$ 420 nm) | n-p | Oxygen vacancies can be assessed by reactions using ethylene glycol as a solvent at a high temperature | [121] |
| Microwave-assisted | $C_3H_6N_6$, $Bi(NO_3)_3·5H_2O$, KCl, $HNO_3$ | BiOCl microplates were grown over the surface of g-$C_3N_4$ nanosheets | Nizatidine (5 ppm) | Mic-LED-365 | Binary heterojunction | pH of the solution was adjusted to match the isoelectric point of the complex materials for enhancing the photocatalytic activity | [122] |
| | | | BiOBr-g-$C_3N_4$ | | | | |
| Reflux process in oil bath | $CH_4N_2O$, $C_2H_6O_2$, $Bi(NO_3)_3·5H_2O$, KBr and $C_2H_5OH$ | BiOBr nanoplates are deposited on the surface of larger g-$C_3N_4$ nanosheets | Rhodamine B (10 ppm) Bisphenol A (5 ppm) | 300 W Xe lamp | Z-scheme | More reactive sites and enhanced mass transfer resulting from larger specific surface area and mesoporosity led to higher activity | [123] |
| Hydrothermal | $C_3H_6N_6$, $C_2H_6O_2$, $Bi(NO_3)_3·5H_2O$ and CTAB | Nanoflakes of g-$C_3N_4$ and BiOBr are observed | Bisphenol A (10 ppm) Methyl orange (10 ppm) Rhodamine B (10 ppm) | 300 W Xe lamp | Binary heterojunction | Surface functional groups of g-$C_3N_4$ provided nucleation sites for reaction by inhibiting the formation of BiOBr assembly | [124] |
| Electrostatic self-assembly | $Bi(NO_3)_3·5H_2O$, KBr, $C_3H_6N_6$, HCl, $CH_3COOH$ and $C_2H_3NaO_2$ | 3D hierarchical flower-like structures of BiOBr are attached to surface of pg-$C_3N_4$ consisting of nanostructures and plicate shapes. | Carbamazepine (5 ppm) | 500 W Xe lamp | Binary heterojunction | Presence of low concentration of bicarbonate accelerated the carbamazepine degradation while nitrate and chloride inhibited its efficiency | [62] |
| Solvothermal | $Bi(NO_3)_3·5H_2O$, CTAB, $C_3H_8O_3$, PVP and $C_3H_6N_6$ | Flower-like microspheres of BiOBr are grown over g-$C_3N_4$ nanosheets | Methyl orange (10 ppm) Rhodamine B (10 ppm) | 500 W Xe lamp | Z-scheme | Optimum content of g-$C_3N_4$ was found to be 5 wt% over the BiOBr nanosheets | [125] |

Table 1. *Cont*.

| Synthesis Method | Precursors | Morphology | Contaminant Parameters | Light Source | Heterojunction Type | Significance of the Result | Ref. |
|---|---|---|---|---|---|---|---|
| Ultrasound-assisted water-bath deposition | $Bi(NO_3)_3 \cdot 5H_2O$, PVP, $C_2H_6O_2$, NaBr, $C_2H_5OH$ and $CH_4N_2O$ | BiOBr nanoflakes are dispersed over the surface of g-$C_3N_4$ nanosheets | Rhodamine B (20 ppm) *E. coli* ($1.0 \times 10^6$ CFU mL$^{-1}$) | Visible light | Z-scheme | Z-scheme photocatalytic mechanism was evidenced from Tafel curve analysis | [99] |
| Solvothermal | $Bi(NO_3)_3 \cdot 5H_2O$, $C_3H_6N_6$, CTAB and $C_2H_6O_2$ | g-$C_3N_4$ nanosheets are compactly combined with BiOBr nanosheets | Rhodamine B (10 ppm) | 500 W Xe lamp | Binary heterojunction | Holes and superoxide radicals played dominant role in the RhB removal | [126] |
| Template-assisted hydrothermal | $Bi(NO_3)_3 \cdot 5H_2O$, NaBr, $C_2H_6O_2$, NaOH, $C_2H_5OH$ and $NH_4Cl$ | BiOBr microspheres are randomly dispersed on the surface of g-$C_3N_4$ | Tetracycline (20 ppm) Rhodamine B (15 ppm) | 500 W Xe lamp | Direct Z-scheme | L-lysine with polar functional groups of amino and hydroxyl, served as bio-template for controlling the crystal growth and self-assembly process of BiOBr | [81] |
| Hydrothermal | $C_3H_6N_6$, $Bi(NO_3)_3 \cdot 5H_2O$, HCl, NaOH, $CH_4N_2O$, $CH_4N_2S$ and KBr | BiOBr nanolayers are distributed on the surface of porous g-$C_3N_4$ nanosheets | Methylene blue (10 ppm) | 50 W 410 nm LED light | Binary heterojunction | Optimized content of Pg-$C_3N_4$ in the binary composite for high activity was found to be 20 wt% | [127] |
| Polycondensation and precipitation | $C_3H_6N_6$, $Bi(NO_3)_3 \cdot 5H_2O$ and CTAB | Mesoporous flower-like BiOBr are grown over porous sheets of g-$C_3N_4$ | Reactive blue 198 (50 ppm) Reactive black 5 (50 ppm) Reactive yellow 145 (50 ppm) | 500 W tungsten lamp | Z-scheme | Degradation pathways were proposed to follow pseudo-first-order kinetics with 30% pGCN-BiOBr | [128] |
| Reflux process | TEOS, $C_2H_5OH$, $NH_4OH$, $C_2H_4N_4$, $NH_4HF_2$, $Bi(NO_3)_3 \cdot 5H_2O$, $CH_4N_2O$ and KBr | BiOBr nanoparticles are uniformly loaded on the surface of IO CN | Levofloxacin (10 ppm) Rhodamine B (20 ppm) | 300 W Xe lamp | Z-scheme | Combination of Z-scheme and inverse opal structure influenced the visible light absorption ability and photocatalytic performance | [129] |
| | | | $Bi_xO_yX_z$-g-$C_3N_4$ | | | | |
| | | | $Bi_7O_9I_3$-g-$C_3N_4$ | | | | |
| Hydrothermal | $Bi(NO_3)_3 \cdot 5H_2O$, $C_3H_6N_6$, NaI, $CN_2H_4S$ and $CH_3COONa$ | Flower-like nanospheres of $Bi_7O_9I_3$ are grown over the surface of g-$C_3N_4$ nanosheets | Doxycycline hydrochloride | Xe lamp | Z-scheme | In situ growth of $Bi_7O_9I_3$ on ultrathin g-$C_3N_4$ via mild and simple hydrothermal means without any toxic reagents | [109] |
| | | | $Bi_5O_7I$-g-$C_3N_4$ | | | | |
| Hydrothermal | $NH_3$, $C_2H_5OH$, TEOS, C18TMOS, $CH_2N_2$, $NH_4HF_2$, $Bi(NO_3)_3 \cdot 5H_2O$, $C_2H_6O_2$ and KI | $Bi_5O_7I$ nanoparticles are grown over surface of porous g-$C_3N_4$ | Phenol (10 ppm) | CEL-HXF300 | p-n | Silica templates were used to obtain lamellar and porous g-$C_3N_4$ in GCN-$Bi_5O_7I$ composite | [130] |

**Table 1.** *Cont.*

| Synthesis Method | Precursors | Morphology | Contaminant Parameters | Light Source | Heterojunction Type | Significance of the Result | Ref. |
|---|---|---|---|---|---|---|---|
| Hydrothermal | Bi(NO₃)₃·5H₂O, KI, C₂H₆O₂ and C₃H₆N₆ | Irregular shaped layers of g-C₃N₄ are covered on the surface of microspheres consisting of self-assembled thin platelets of Bi₅O₇I | Methyl orange (10 ppm) Rhodamine B (10 ppm) | 300 W Xe lamp | Z-scheme Heterojunction | g-C₃N₄-Bi₅O₇I-10 showed better performance towards dye degradation in acidic conditions | [107] |
| Hydrolysis and thermal condensation | C₂H₄N₄ Bi(NO₃)₃·4H₂O, C₂H₆O₂, NaOH, HCl, C₆H₁₅N, KH₂PO₄ and KI | Rod-like patterns of Bi₅O₇I are embedded on D-g-C₃N₄ | Metronidazole (15 ppm) | 300 W Xe lamp | Binary heterojunction | Charge carrier separation in the composite was evidenced from photocurrent response measurements | [131] |
| | | | Bi₄O₅I₂-g-C₃N₄ | | | | |
| Mixed calcination | C₃H₆N₆ Bi(NO₃)₃·4H₂O, NaOH and KI | Bi₄O₅I₂ nanoflakes are grown on g-C₃N₄ nanosheets | Rhodamine B (1 × 10⁻⁵ M) NO removal | 300 W tungsten halogen lamp | n-n | Super oxide radicals and holes are active species during the degradation | [132] |
| Ionic liquid-assisted solvothermal | C₆H₁₄O₆, C₁₀H₁₉IN₂, Bi(NO₃)₃·4H₂O, NaOH and KI | Bi₄O₅I₂ nanosheets are dispersed on g-C₃N₄ nanosheets | Rhodamine B (10 ppm) Bisphenol A (10 ppm) | 300 W Xe lamp | Binary hetrojunction | [Hmim]I played multiple roles during the synthesis which was propitious for heterojunction formation | [87] |
| Hydrothermal and heating | C₃H₆N₆, Bi(NO₃)₃·4H₂O, C₂H₆O₂ and KI | Hierarchical microspheres of Bi₄O₅I₂ are grown on the surface of g-C₃N₄ | Methyl orange (20 ppm) | 350 W Xe lamp | Type-II | The in situ transformation endowed the composite with good contact between the semiconductors in construction of tight heterojunction | [110] |
| | | | Bi₃O₄Cl-g-C₃N₄ | | | | |
| Mixing and heating | Bi(NO₃)₃·5H₂O, HCl, Bi₂O₃ and C₃H₆N₆ | Irregular blocks consisting of a lot of nanoflakes of Bi₃O₄Cl attached onto the surface of g-C₃N₄ | Rhodamine B (10 ppm) | 350 W Xe lamp | Binary heterojunction | Coupling Bi₃O₄Cl on g-C₃N₄ improved the specific surface area and charge carrier separation | [133] |
| Solid phase calcination | Bi(NO₃)₃·5H₂O, CH₄N₂O, NH₄Cl and C₂H₆O₂ | Bi₃O₄Cl nanoflakes are grown on g-C₃N₄ nanosheets | Rhodamine B (10 ppm) Tetracycline (10 ppm) Hexavalent chromium (10 ppm) | 250 W Xe lamp | Z-scheme | Shorter fluorescent lifetime (0.952 ns) attributed to additional nonradioactive decay channel for electron transfer from Bi₃O₄Cl to g-C₃N₄ | [96] |
| | | | Bi₁₂O₁₇Cl₂-g-C₃N₄ | | | | |
| Chemical precipitation | CH₄N₂O, BiCl₃, C₂H₅OH and NaOH | Bi₁₂O₁₇Cl₂ nanosheets are grown on surface of g-C₃N₄ | Rhodamine B (5 ppm) Methyl orange (10 ppm) | 300 W Xe lamp | Binary heterojunction | Hydroxyl radicals and holes main active species during the reaction as evidenced from electron spin resonance technique | [60] |

**Table 1.** *Cont.*

| Synthesis Method | Precursors | Morphology | Contaminant Parameters | Light Source | Heterojunction Type | Significance of the Result | Ref. |
|---|---|---|---|---|---|---|---|
| | | | $Bi_4O_5Br_2$-g-$C_3N_4$ | | | | |
| Ionic liquid-assisted solvothermal | $[C_{16}mim]Br$, $C_3N_3(NH_2)_3$, $C_2H_3N$, $C_3N_3Cl_3$, $Bi(NO_3)_3 \cdot 5H_2O$, $C_6H_{14}O_6$ and NaOH | Rod-like g-$C_3N_4$ has closely combined with sheet-like $Bi_4O_5Br_2$ | Ciprofloxacin (10 ppm) Rhodamine B (10 ppm) | 300 W Xe lamp | Binary heterojunction | Ionic liquid $[C_{16}mim]Br$ served as solvent, dispersing agent and reactant for the distribution of $Bi_4O_5Br_2$ over g-$C_3N_4$ | [134] |
| Precipitation | $C_3H_6N_6$, $C_2H_5OH$, $BiBr_3$ and NaOH | Irregular nanosheets of $Bi_4O_5Br_2$ were stacked with g-$C_3N_4$ sheets | Rhodamine B (10 ppm) Tetracycline (10 ppm) | 72 W LED lamp | Binary heterojunction | Improved adsorptive nature in BBO/CN-75 is due to generation of more Lewis base sites as confirmed by Zeta potential studies | [135] |
| Solvothermal | $C_3H_6N_6$, $Bi(NO_3)_3 \cdot 5H_2O$, $[C_{16}mim]Br$, $C_6H_{14}O_6$, NaOH and $C_2H_5OH$ | Ultrathin $Bi_4O_5Br_2$ nanosheets are dispersed on the graphene-like g-$C_3N_4$ nanosheets | Ciprofloxacin (10 ppm) Rhodamine B (10 ppm) | 300 W Xe lamp | Binary heterojunction | Red shift in the bandgap absorption was observed with introduction of graphene-like g-$C_3N_4$ | [85] |
| | | | Noble metal coupled BiOX-g-$C_3N_4$ | | | | |
| | | | BiOI/Pt/g-$C_3N_4$ | | | | |
| Two-step (reduction and stirring) | $CH_4N_2O$, $NaBH_4$, $H_2PtCl_6 \cdot 6H_2O$, $C_2H_6O_2$, $Bi(NO_3)_3 \cdot 5H_2O$ and KI | Pt nanoparticles and BiOI hierarchical structure grew on the g-$C_3N_4$ sheets | Phenol (25 ppm) Tetracycline hydrochloride (20 ppm) | Visible light | Solid-state Z-scheme | Unobstructed Z-scheme charge carrier transfer pathways in BiOI/Pt/g-$C_3N_4$ composite are discussed in relevance to phenol and tetracycline oxidation | [136] |
| | | | g-$C_3N_4$/Eu/$Bi_{24}O_{31}Cl_{10}$ (BOC) | | | | |
| Impregnation-calcination | $Bi(NO_3)_3 \cdot 5H_2O$, $C_3H_6N_6$, $NH_4Cl$, $C_6H_8O_7$, $HNO_3$, $NH_3 \cdot H_2O$ and $Eu(NO_3)_3 . 6H_2O$ | g-$C_3N_4$ nanosheets were coated on the surface of irregular shaped smaller sized crystal particles of Eu-doped BOC | Rhodamine B (10 ppm) | 250 W Xe lamp | Binary heterojunction | CN/Eu-BOC exhibited higher performance than CN/BOC suggesting that Eu (III) could be used as cocatalyst | [137] |
| | | | g-$C_3N_4$/Au/BiOBr | | | | |
| Hydrothermal and in situ reduction | $C_3H_6N_6$, $Bi(NO_3)_3 \cdot 5H_2O$, KBr, $C_2H_5OH$, $C_8H_{11}NO_2$ and $HAuCl_4 \cdot 4H_2O$ | Au nanoparticles are decorated over the surface of lamellar structure of g-$C_3N_4$ and BiOBr sheets | Phenol (10 ppm) | 300 W Xe lamp | Plasmonic Z-scheme | Strong surface plasmon resonance caused by Au NPs contributed to extension of visible light absorption in the ternary composite | [138] |
| Chemical reduction | $Bi(NO_3)_3 \cdot 5H_2O$, KBr, $CH_4N_2S$, CTAB, $Na_3C_6H_5O_7$, $AuCl_3$ and $C_2H_5OH$ | Au nanoparticles were uniformly distributed over the surface of g-$C_3N_4$/BiOBr | Rhodamine B (10 ppm) $CO_2$ reduction | 300 W Xe lamp | Surface plasmon resonance and Z-scheme | Correlation between size of Au NPs and wavelength dependent photocatalytic activity associated with Au-GCN-BiOBr composite is described | [139] |

**Table 1.** *Cont*.

| Synthesis Method | Precursors | Morphology | Contaminant Parameters | Light Source | Heterojunction Type | Significance of the Result | Ref. |
|---|---|---|---|---|---|---|---|
| Carbon material coupled BiOX-g-C$_3$N$_4$ | | | | | | | |
| g-C$_3$N$_4$/CDs/BiOI | | | | | | | |
| Precipitation | CN$_2$H$_2$, C$_3$H$_6$N$_6$, C$_6$H$_8$O$_7$, Bi(NO$_3$)$_3$·4H$_2$O and NaI | CDs and BiOI nanoparticles are grown in intimate contact with gCN nanosheets | Rhodamine B (2.5 × 10$^{-5}$ M) Methylene blue (2.5 × 10$^{-5}$ M) Methyl orange (2.5 × 10$^{-5}$ M) Fuchsine (9.20 × 10$^{-6}$ M) | 50 W LED lamp | Ternary heterojunction | Co-operative effects of CQDs and g-C$_3$N$_4$ promoted the activity of BiOI towards the degradation of organic dyes | [140] |
| GO/g-C$_3$N$_4$/BiOI | | | | | | | |
| In situ generation | CH$_4$N$_2$O, Bi(NO$_3$)$_3$·4H$_2$O, C$_2$H$_6$O$_2$, KI and GO | Flower-like BiOI nanosheets are overlapped with lamellar structure of CN and sheet-like GO. | Methyl orange (10 ppm) Tetracycline (20 ppm) E. coli (50 ppm) S. aureus (50 ppm) | LED lamp | Ternary heterojunction | Loading GO over CN/BiOI resulted in double-charge-transfer at the interface | [141] |
| g-C$_3$N$_4$/MCNTs/BiOI | | | | | | | |
| Solvothermal | CN$_2$H$_2$, KI, Bi(NO$_3$)$_3$·4H$_2$O and C$_2$H$_6$O$_2$ | BiOI nanoparticles are uniformly loaded on surface of g-C$_3$N$_4$- MCNTs. | Methylene blue (10 ppm) | 300 W Xe lamp | Z-scheme | MCNTs facilitated the electron transfer from BiOI to g-C$_3$N$_4$ resulting in Z-scheme charge transfer pathway | [142] |
| g-C$_3$N$_4$/BiOI/rGO immobilized on Ni foam | | | | | | | |
| Hydrothermal and reduction | CH$_4$N$_2$O, C$_2$H$_4$N$_4$, NaSO$_4$, Bi(NO$_3$)$_3$·5H$_2$O, C$_2$H$_6$O$_2$, C$_2$H$_6$O, NH$_3$, H$_4$N$_2$·H$_2$O, Ni foam, GO and NaI | Laminar structures of g-C$_3$N$_4$, BiOI and sheet-like rGO form the ternary sheet-like hybrids and are immobilized on the surface of Ni foam. | Methyl orange (5 ppm) CO$_2$ | 300 W Xe lamp | Hybrid Z-scheme | rGO functioned as both electron mediator and binder while Ni foam improved the reusability of the composite | [143] |
| g-C$_3$N$_4$/CDs/BiOCl | | | | | | | |
| Refluxing | C$_3$H$_6$N$_6$, Bi(NO$_3$)$_3$·5H$_2$O, NaCl, CH$_4$N$_2$O and C$_6$H$_8$O$_7$ | Smaller spherical particles of CDs and rod-like particles of BiOCl are grown on the surface of g-C$_3$N$_4$ nanosheets | Rhodamine B (1 × 10$^{-5}$ M) Methylene blue (1 × 10$^{-5}$ M) Methyl orange (1 × 10$^{-5}$ M) Fuchsine (0.77 × 10$^{-5}$ M) Phenol (5 × 10$^{-5}$ M) | 50 W LED lamp | Ternary heterojunction | Formation of g-C$_3$N$_4$/CDs/BiOCl composite influenced the optical properties and photocatalytic performance | [93] |
| BiOBr/rGO/pg-C$_3$N$_4$ | | | | | | | |
| Solvothermal | CH$_4$N$_2$O, HCl, rGO, C$_2$H$_6$O$_2$, Bi(NO$_3$)$_3$·5H$_2$O and CTAB | BiOBr and rGO nanosheets are dispersed simultaneously on the surface of pg-C$_3$N$_4$ | Rhodamine B (10 ppm) Tetracycline (10 ppm) | 300 W Xe lamp | Ternary Z-scheme | Optimized content of BiOBr in ternary composite for high activity was found to be 10 wt% | [144] |

**Table 1.** *Cont.*

| Synthesis Method | Precursors | Morphology | Contaminant Parameters | Light Source | Heterojunction Type | Significance of the Result | Ref. |
|---|---|---|---|---|---|---|---|
| | | | BiOBr/CDs/g-C$_3$N$_4$ | | | | |
| Hydrothermal | Bi(NO$_3$)$_3$·5H$_2$O, KBr, HNO$_3$, CH$_4$N$_2$O, NH$_4$Cl, C$_6$H$_8$O$_7$ and C$_2$H$_4$N$_4$ | Ultrathin nanosheets | Ciprofloxacin (10 ppm) Tetracycline (20 ppm) | 300 W Xe lamp | Z-scheme | Up-converted PL character and short charge transport distance of CDs were beneficial towards broadened light absorption and remarkable interfacial charge transfer | [145] |
| | | | CNNs/CDs/BiOBr | | | | |
| Refluxing | C$_3$H$_6$N$_6$, C$_6$H$_8$O$_7$, CH$_4$N$_2$O, Bi(NO$_3$)$_3$·5H$_2$O and NaBr | CDs and BiOBr nanoparticles are accumulated on the surface of carbon nitride nanosheets (CNNs) | Rhodamine B (1 × 10$^{-5}$ M) Methylene blue (1 × 10$^{-5}$ M) Methyl orange (1 × 10$^{-5}$ M) Cr(VI) (100 ppm) | 50 W LED lamp | Ternary Z-scheme | CNNs/CDs/BiOBr was stable even after five consecutive cycles towards the degradation of pollutants with fresh dye solution each time | [94] |
| | | | g-C$_3$N$_4$/BiOBr-rGO | | | | |
| Two-step hydrothermal assembly route | Graphite powder, C$_8$H$_{11}$NO$_2$, H$_2$SO$_4$, HNO$_3$, KMnO$_4$, H$_2$O$_2$, Bi(NO$_3$)$_3$·5H$_2$O, KBr and C$_3$H$_6$N$_6$ | Flake-like BiOBr are covered by thin layer of g-C$_3$N$_4$ film | Rhodamine B (10 ppm) | 300 W Xe lamp | p-n | Immobilization of the powder catalyst on 3D RGO aerogel surface collectively contributed to excellent recycling process of the catalyst | [146] |
| | | | Carbon Fibers/g-C$_3$N$_4$/BiOBr | | | | |
| Chemical bath deposition | Carbon fibres, CH$_4$N$_2$O, Bi(NO$_3$)$_3$·5H$_2$O, C$_4$H$_9$NO and KBr | Growth of g-C$_3$N$_4$ nanosheets and BiOBr nanoplates on carbon fibers (CFs) | Tetracycline (20 ppm) | 300 W Xe lamp | Ternary heterojunction | Recyclable cloth-shaped CFs/g-C$_3$N$_4$/BiOBr bundles had great mechanical strength | [147] |
| | | | BiOBr/CS/g-C$_3$N$_4$ | | | | |
| Solvothermal | C$_3$H$_6$N$_6$, C$_6$H$_{12}$O$_6$, Bi(NO$_3$)$_3$·5H$_2$O, KBr and C$_2$H$_6$O$_2$ | Spherical carbon spheres are wrapped uniformly with g-C$_3$N$_4$ and BiOBr matrix | Rhodamine B (10 ppm) | 300 W W halogen lamp | Ternary heterojunction | Carbon spheres were used as interlinking network between g-C$_3$N$_4$ and BiOBr matrix for effective electron transfer | [148] |
| | | | BiOI/porous g-C$_3$N$_4$/graphene hydrogel | | | | |
| Hydrothermal | Bi(NO$_3$)$_3$·5H$_2$O, KI, CH$_4$N$_2$O, C$_2$H$_6$O$_2$, | BiOI and porous g-C$_3$N$_4$ were loaded onto 3D cross-linking graphene hydrogel | Methylene blue (40 ppm) Levofloxacin (20 ppm) | 300 W Xe lamp | Ternary heterojunction | 3D graphene hydrogel played multiple roles: enhanced adsorption ability, provided bulk electron transfer channels, rendered easy separation and recycling | [149] |

**Table 1.** *Cont.*

| Synthesis Method | Precursors | Morphology | Contaminant Parameters | Light Source | Heterojunction Type | Significance of the Result | Ref. |
|---|---|---|---|---|---|---|---|
| | | | Semiconductor coupled BiOX-g-C$_3$N$_4$ | | | | |
| | | | Polyacrylonitrile/g-C$_3$N$_4$/BiOI nanofibres | | | | |
| Impregnation | Bi(NO$_3$)$_3$·5H$_2$O, KI, C$_3$H$_6$N$_6$, N,N-dimethylformamide (C$_3$H$_7$NO) and polyacrylonitrile (C$_3$H$_3$N)$_n$ | BiOI nanostructures are uniformly dispersed over PAN/g-C$_3$N$_4$ nanofibres | Rhodamine B (10 ppm) Cr (VI) (20 ppm) | 300 W Xe lamp | - | Ultralong 1D macroscopic flexible self-supporting floating structures prevented agglomeration and loss of catalyst during recycling | [150] |
| | | | SiO$_2$@g-C$_3$N$_4$/BiOI nanofibres | | | | |
| Impregnation | Polyvinylpyrrolidone (C$_6$H$_9$NO)$_n$, TEOS (SiC$_8$H$_{20}$O$_4$), ethanol, C$_3$H$_6$N$_6$, Bi(NO$_3$)$_3$·5H$_2$O and KI | BiOI nanosheets are loaded on the surface of ultrathin g-C$_3$N$_4$@SiO$_2$ nanofibres. | Rhodamine B (10 ppm) | 150 W Xe lamp | Direct Z-scheme | Depositing SiO$_2$ NFs at BiOI/g-C$_3$N$_4$ interface improved Z-scheme charge carrier separation and recyclability | [151] |
| | | | BiOI/AgI/g-C$_3$N$_4$ | | | | |
| In situ crystallization | Bi(NO$_3$)$_3$·5H$_2$O, KI, AgNO$_3$ and C$_3$H$_6$N$_6$ | Irregular nanoparticles of AgI are grown on the surface of g-C$_3$N$_4$ covered with BiOI nanoflakes. | Methyl orange (10 ppm) Cr(VI) (50 ppm) | 300 W Xe lamp | Ternary heterojunction | Visible light response was tailored from 460 to 560 nm by increasing the content of AgI in the composite | [152] |
| | | | BiOI/g-C$_3$N$_4$/CeO$_2$ | | | | |
| Calcination and hydrothermal | Bi(NO$_3$)$_3$·5H$_2$O, KI, Ce(NO$_3$)$_3$.6H$_2$O and C$_3$H$_6$N$_6$ | BiOI microspheres and CeO$_2$ nanoparticles are randomly adhered to the surface of g-C$_3$N$_4$ | Tetracycline (20 ppm) | 300 W Xe lamp | Ternary heterojunction | Optimum content of CeO$_2$ in the ternary hybrid was found to be 3 wt% towards efficient TC degradation | [153] |
| | | | BiOI@MIL-88A(Fe)@g-C$_3$N$_4$ | | | | |
| Hydrothermal | Bi(NO$_3$)$_3$·5H$_2$O, KI, C$_2$H$_6$O$_2$, C$_3$H$_6$N$_6$, FeCl$_3$·6H$_2$O and C$_4$H$_4$O$_4$ | BiOI flower-like hierarchical microspheres are loaded on the surface of MIL-88A(Fe)@g-C$_3$N$_4$ with core@shell structure | Acid blue 92 (10 ppm) Rhodamine B (10 ppm) Phenol (10 ppm) | 300 W Xe lamp | Ternary heterojunction | g-C$_3$N$_4$ deposited over BiOI@MIL-88A(Fe) via hydrothermal method facilitated carrier separation in the composite | [154] |
| | | | g-C$_3$N$_4$/Fe$_3$O$_4$/BiOI | | | | |
| Reflux and precipitation | Bi(NO$_3$)$_3$·4H$_2$O, NaI, C$_3$H$_6$N$_6$, FeCl$_3$·6H$_2$O, FeCl$_2$·4H$_2$O and NH$_3$ | Fe$_3$O$_4$ particles and BiOI are grown on the surface of g-C$_3$N$_4$ sheets | Rhodamine B (1 × 10$^{-5}$ M) Methylene blue (1.3 × 10$^{-5}$ M) Methyl orange (1.05 × 10$^{-5}$ M) | 50 W LED source | Ternary heterojunction | g-C$_3$N$_4$/Fe$_3$O$_4$/BiOI was magnetically separated from the aqueous medium within a short span of time | [92] |
| | | | g-C$_3$N$_4$/I$^{3-}$-BiOI | | | | |
| Solvothermal | C$_3$H$_6$N$_6$, Bi(NO$_3$)$_3$·4H$_2$O, C$_4$H$_6$O$_6$, C$_4$H$_{10}$O, EDTA-2Na, C$_6$H$_8$O$_6$, K$_2$Cr$_2$O$_7$, NaN$_3$, DMPO and DMSO | Flower-like microspheres containing ultrathin nanosheets of BiOI are loaded g-C$_3$N$_4$ | Methyl mercaptan (CH$_3$SH) (70 ppm) | 8 W LED | Z-scheme | CH$_3$SH removal monitored via in situ DRIFTS and the intermediate and conversion pathways were elucidated | [63] |

**Table 1.** *Cont.*

| Synthesis Method | Precursors | Morphology | Contaminant Parameters | Light Source | Heterojunction Type | Significance of the Result | Ref. |
|---|---|---|---|---|---|---|---|
| | | | $MoS_2/g\text{-}C_3N_4/Bi_{24}O_{31}Cl_{10}$ | | | | |
| Impregnation-calcination | $Bi(NO_3)_3\cdot 5H_2O$, $C_3H_6N_6$, $NH_4Cl$, $C_6H_8O_7$, $HNO_3$, $NH_3\cdot H_2O$, $(NH_4)_6Mo_7O_{24}.4H_2O$ and DMF | Numerous g-$C_3N_4$ nanosheets and flower-like $MoS_2$ are grown and combined with irregular block-like shapes of BOC | Tetracycline (20 ppm) | 300 W Xe lamp | Dual Z-scheme ternary heterojunction | Carrier lifetime was higher in CN/MS/BOC (3.9782 ns) compared to BOC (1.0163 ns) | [155] |
| | | | $BiOCl/Bi_2MoO_6/g\text{-}C_3N_4$ | | | | |
| Refluxing | $Bi(NO_3)_3\cdot 5H_2O$, $Na_2MoO_4\cdot 2H_2$, HCl and NaOH | Combination of irregular rodlike, platelet-shaped and sheet-shaped morphologies | Rhodamine B (0.5 mM) | 350 W Xe lamp | Ternary heterojunction | $BiOCl/Bi_2MoO_6$ immobilized on g-$C_3N_4$ surface exhibited dual functionality as photocatalysts and optical limiters | [156] |
| | | | $BiOCl/CdS/g\text{-}C_3N_4$ | | | | |
| Solvothermal cum co-precipitation | $C_3H_6N_6$, HCl, $C_4H_6CdO_4$, DMSO, $Bi(NO_3)_3\cdot 5H_2O$, $C_3H_8O$ and NaOH | Growth of hierarchical BiOCl nanoflowers with embedded CdS nanoparticles on g-$C_3N_4$ nanosheets | Rhodamine B (20 ppm) Phenol | 400 W Ne-illuminator | Ternary heterojunction | Presence of two visible light active components led to highly efficient electron transfer in multicomponent heterojunction | [157] |
| | | | $BiOCl/g\text{-}C_3N_4/kaolinite$ | | | | |
| Two-step layer-by-layer self-assembly | $C_2H_4N_4$, CTAC, $Bi(NO_3)_3\cdot 5H_2O$, $CH_3COOH$, HCHO and kaolinite | g-$C_3N_4$ and BiOCl ultrathin nanosheets are covered on the surface of kaolinite lamellar with single layer | Rhodamine B (10 ppm) | 500 W Xe lamp | Ternary heterojunction | Holes dominated the degradation pathways for BiOCl/g-$C_3N_4$/kaolinite | [158] |
| | | | $g\text{-}C_3N_4/g\text{-}C_3N_4/BiOBr$ | | | | |
| Thermal de-composition and solvother-mal | $CH_4N_2O$, $CH_4N_2S$, $Bi(NO_3)_3\cdot 5H_2O$, KBr, $NaC_{12}H_{25}SO_4$ and $C_2H_6O_2$ | g-$C_3N_4$ prepared using thiourea-urea complex was uniformly dispersed on the nanosheets of the flower-like BiOBr | Rhodamine B (20 ppm) Fluorescein isothiocyanate (20 ppm) Tetracycline hydrochloride (20 ppm) | High pressure Xe lamp | Ternary direct Z-scheme + isotype heterojunction | Combined effect of Z-scheme + isotype heterojunction charge transfer pathways was observed in g-$C_3N_4$/g-$C_3N_4$/BiOBr | [159] |
| | | | $AgBr/g\text{-}C_3N_4/BiOBr$ | | | | |
| Hydrothermal and in situ ion-exchange | $CH_4N_2O$, KBr, $Bi(NO_3)_3\cdot 5H_2O$, $AgNO_3$ and $C_2H_6O_2$ | AgBr nanoparticles are dispersed on the surface of g-$C_3N_4$/BiOBr nanosheets | Rhodamine B (10 ppm) Tetracycline (10 ppm) | 300 W Xe lamp | Ternary heterojunction | Influence of AgBr loading on GCN/BOB composite towards the photocatalytic activity is discussed in detail | [160] |
| | | | $Brookite/g\text{-}C_3N_4/BiOBr$ | | | | |
| Hydrothermal | $TiCl_4$, $Bi(NO_3)_3\cdot 5H_2O$, $C_3H_6N_6$, CTAB, $NH_4Cl$, KBr and NaOH | Spindle shaped brookite are wrapped by the layer structure of BiOBr which are further wrapped by lamellar g-$C_3N_4$ | Rhodamine B (10 ppm) | 70 W metal halide lamp | Ternary heterojunction | Ternary composite had ability to destroy the oxygen heteroanthracene ring and chromogenic group of RhB | [161] |

**Table 1.** *Cont.*

| Synthesis Method | Precursors | Morphology | Contaminant Parameters | Light Source | Heterojunction Type | Significance of the Result | Ref. |
|---|---|---|---|---|---|---|---|
| | | | BiOCl/g-C$_3$N$_4$@UiO-66 | | | | |
| Solvothermal | Bi(NO$_3$)$_3$·5H$_2$O, KCl, CH$_3$COOH, C$_3$H$_6$N$_6$, ZrCl$_4$, C$_8$H$_6$O$_4$, C$_3$H$_7$NO | BiOCl nanoplates and g-C$_3$N$_4$ nanosheets were decorated over the surface of UiO-66 | Rhodamine B (10 ppm) | 250 W Xe lamp | Ternary heterojunction | UiO-66 was proved beneficial to the photocatalytic reaction by enlarging the photoadsorption and preventing the electron-hole recombination | [121] |
| | | | g-C$_3$N$_4$/BiOI/Bi$_2$O$_2$CO$_3$ | | | | |
| Simple reflux and in situ ion exchange | Bi(NO$_3$)$_3$·5H$_2$O, C$_3$H$_6$N$_6$, KI, NaHCO$_3$, Na$_2$SO$_4$, CH$_3$CH$_2$OH, Kr$_2$Cr$_2$O$_7$, H$_2$SO$_4$, H$_3$PO$_4$ | Thin nanosheets of BiOI are distributed over g-C$_3$N$_4$ layers | Rhodamine B (10 ppm) | 250 W Xe lamp | Ternary heterojunction | Based on the matched energy levels, BiOI acted as the charge transmission bridge | [162] |
| | | | BiOX and BiOY coupled g-C$_3$N$_4$ | | | | |
| | | | Bi$_7$O$_9$I$_3$/Bi$_5$O$_7$I/g-C$_3$N$_4$ | | | | |
| Hydrothermal | C$_3$H$_6$N$_6$, Bi(NO$_3$)$_3$·4H$_2$O, C$_2$H$_6$O$_2$ and KI | Irregular rods consisting of thin irregular nanosheets | Crystal violet (10 ppm) | 150 W Xe lamp | Binary heterojunction | Controlled synthesis of series of BiO$_x$I$_y$/g-C$_3$N$_4$ composites is reported | [163] |
| | | | g-C$_3$N$_4$/BiOI/BiOBr | | | | |
| Chemical precipitation | CH$_4$N$_2$O, C$_2$H$_6$O$_2$, Bi(NO$_3$)$_3$·5H$_2$O, KBr and KI | Curved g-C$_3$N$_4$ nanosheets are attached to the surface of BiOI/BiOBr exhibiting sphere-like structures containing thin nanosheets of BiOI on large plates of BiOBr | Methyl orange (10 ppm) Escherichia coli (ATCC 15597) | 300 W Xe lamp | Ternary heterojunction | Presence of BiOI shifted the bandgap to longer wavelength and also suppressed the carrier recombination | [164] |
| | | | g-C$_3$N$_4$@BiOCl/Bi$_{12}$O$_{17}$Cl$_2$ | | | | |
| In situ self-assembly | CH$_4$N$_2$O, BiCl$_3$, C$_2$H$_5$OH and NaOH | Combination of layered and irregular microstructures having smooth nanosheets of different sizes are grown over g-C$_3$N$_4$ | NO removal (1 × 10$^{-9}$ ppb) | 100 W commercial tungsten halogen lamp | Ternary heterojuncton | Electron spin resonance proved that both hydroxyl and superoxide radicals are active species towards NO removal | [165] |
| | | | BiOI/BiOCl/g-C$_3$N$_4$ | | | | |
| Precipitation | Bi(NO$_3$)$_3$·5H$_2$O, CH$_4$N$_2$O, KI, KCl and NH$_3$ | Nanosheets are stacked densely to form irregular microstructures over thin layers of g-C$_3$N$_4$ | Acid orange (10 ppm) | 400 W Halogen lamp | Ternary heterojuncton | The optimal ratio of ternary hybrid was found to be 5:3:2 | [91] |

**Table 1.** *Cont.*

| Synthesis Method | Precursors | Morphology | Contaminant Parameters | Light Source | Heterojunction Type | Significance of the Result | Ref. |
|---|---|---|---|---|---|---|---|
| | | | Quaternary heterojunction | | | | |
| | | | $BiOCl/g\text{-}C_3N_4/Cu_2O/Fe_3O_4$ | | | | |
| Co-precipitation | $Bi(NO_3)_3 \cdot 5H_2O$, KCl, NaOH, $HNO_3$, $CuSO_4 \cdot 5H_2O$, $C_2H_5OH$, $FeCl_3$, $FeCl_2$ $CH_4N_2S$ and $C_2H_6O_2$ | Flower shaped BiOCl, spherical $Fe_3O_4$ and cubical $Cu_2O$ nanoparticles are connected with porous sheets of $g\text{-}C_3N_4$ | Sulfamethoxazole (100 µM) | 800 W Xe lamp Natural sunlight | Quaternary nano-heterojunction | p-n-p junction functioned well under both artificial visible light and solar light towards sulfamethoxazole degradation | [166] |
| | | | $g\text{-}C_3N_4/BiOI/BiOBr$ | | | | |
| Solvothermal | $C_2H_4N_4$, $C_2H_6O_2$, $Bi(NO_3)_3 \cdot 5H_2O$, KI and CTAB | $g\text{-}C_3N_4$ was attached to the surface of quadrate BiOBr substrates overlapped with rounded thin pieces of the BiOI | Methylene blue (20 ppm) | 500 W Xe lamp | Ternary Z-scheme | Charge carrier dynamics in ternary composite is reviewed based on transient photocurrent response | [167] |
| | | | Doped $BiOX\text{-}g\text{-}C_3N_4$ | | | | |
| | | | K-doped $g\text{-}C_3N_4/BiOBr$ | | | | |
| In situ synthesis | $CH_4N_2O$, CTAB, KOH and $Bi(NO_3)_3 \cdot 5H_2O$, | 2D nanosheets | Rhodamine B (20 ppm) Tetracycline (10 ppm) | 500 W Xe lamp | Binary heterojunction | K was interfaced with $g\text{-}C_3N_4/BiOBr$ for improved migration and transportation of photogenic carriers | [168] |
| | | | $g\text{-}C_3N_4@Bi/BiOBr$ | | | | |
| Solvothermal | $C_3H_6N_6$, $Bi(NO_3)_3 \cdot 5H_2O$, $C_2H_6O_2$, KBr and $C_2H_5OH$ | 3D fluffy and hierarchical structure where Bi/BiOBr nanoplates are embedded on the surface of the layered $g\text{-}C_3N_4$ | Rhodamine B (20 ppm) Tetracycline (12 ppm) | Simulated sunlight | Ternary indirect Z-scheme | Ethylene glycol functioned as solvent and a reductant for tuning the morphology and boosting the photocatalytic performance | [84] |
| | | | $g\text{-}C_3N_4@Polydopamine/BiOBr$ | | | | |
| Solvothermal | $C_3H_6N_6$, HCl, Da.HCl, NaOH, $Bi(NO_3)_3 \cdot 5H_2O$, PVP, $C_2H_6O_2$ and KBr | Flower-like BiOBr are deposited on the surface of sheet-like $g\text{-}C_3N_4@PDA$ | Sulfamethoxazole (2.5 ppm) | 300 W Xe lamp | Z-scheme | Biomimetic PDA as electron transfer mediator bridging $g\text{-}C_3N_4$-BiOBr was reported for the first time | [169] |

## 4.2. Carbon Dioxide Reduction

The rapid increase in the concentration of atmospheric $CO_2$ as a green-house gas has drawn significant concerns over its huge impact on the global climate. Therefore, the photocatalytic reduction of $CO_2$ to value-added chemicals such as CO, $CH_3OH$, HCOOH, $CH_4$, etc., under direct solar irradiation is pivotal for not only reducing the level of atmospheric $CO_2$, but also for partly fulfilling the renewable fuel demand that may increase in the future, partly owing to the steadily depleting fossil fuel reserves and also due to our environmental policy on curbing the usage of fossil fuels for inhibiting $CO_2$ emission. As mentioned earlier, BiOX photocatalysts are mainly employed for the photocatalytic degradation of organic pollutants and are seldom effective in the reduction of $CO_2$ conversion at neutral condition due to its positive CB position [170]. Therefore, only a few photocatalytic reduction reactions of pristine BiOX for photocatalytic $CO_2$ conversion have been reported

to date [90,171–179]. On the other hand, theoretical studies indicated that the increase in the Bi-content in BiOX could promote the reduction power of photogenerated electrons and increase the thermodynamic force for initiating many reduction reactions that were not possible to be carried out with BiOX. In this regard, non-stoichiometric $Bi_xO_yX_z$ photocatalysts were found to exhibit promising potential in the photoreduction of $CO_2$ to solar fuels and exhibited good stability and possessed suitable band structures for extended visible light absorption with negative CB positions [180]. For instance, Ye et al. reported an enhanced rate of CO and $CH_4$ generation by the photocatalytic reduction of $CO_2$ using $Bi_4O_5Br_2$ microspheres assembled with ultrathin nanosheets in comparison to BiOBr with ultrathin nanosheets and bulk BiOBr. It was proved that Bi-rich $Bi_4O_5Br_2$ with a more negative CB position exhibited enhanced photoreduction of $CO_2$ in comparison to BiOBr. Further, it was revealed that the ultrathin nanosheet morphology of both $Bi_4O_5Br_2$ and BiOBr considerably reduced the recombination due to IEF generation and supported the generation of CO in comparison to bulk BiOBr [181]. Similarly, ultrathin $Bi_4O_5Br_2$ nanosheets synthesized through the molecular precursor method exhibited enhanced performance towards $CO_2$ reduction under visible light irradiation in comparison to bulk $Bi_4O_5Br_2$. The amount of $CO_2$ converted to CO was 63.13 $\mu$mol g$^{-1}$ using $Bi_4O_5Br_2$ ultrathin nanosheets, which was ~2.3 times greater than that of bulk of $Bi_4O_5Br_2$ (27.56 $\mu$mol g$^{-1}$) [182]. The reason for the enhanced $CO_2$ reduction ability using $Bi_4O_5Br_2$ ultrathin nanosheets in comparison to its bulk counterpart was attributed to porous architecture with larger surface area, more negative CB position (−1.19 V), lower rate of recombination of the photogenerated charge carriers and higher photocurrent response. In another study, $Bi_4O_5I_2$ and $Bi_5O_7I$ photocatalysts were successfully synthesized via hydrolyzation and calcination, respectively, using the molecular precursor method. Both $Bi_4O_5I_2$ and $Bi_5O_7I$ exhibited the photocatalytic reduction of $CO_2$ to selectively generate CO, but the higher CB edge and lower bandgap energy (2.18 eV) of $Bi_4O_5I_2$ enabled it to exhibit enhanced photocatalytic performance that was ~11.5 and ~28.3 times greater than that of $Bi_5O_7I$ and BiOI, respectively [183]. In addition to Bi-rich strategy, the hybridization of $Bi_4O_5I_2$ with g-$C_3N_4$ was employed for enhancing the photoreduction of $CO_2$ by the formation of a heterojunction with an $I_3^-/I^-$ redox mediator synthesized through the complex precursor method. The composite exhibited higher photocatalytic activity for $CO_2$ conversion than pure g-$C_3N_4$ and $Bi_4O_5I_2$ owing to the $I_3^-/I^-$ redox mediator formed in situ, which assisted the transfer of the photogenerated charge carriers through the Z-scheme heterojunction and suppressed their recombination [184]. The amount of CO generated by the photocatalytic reduction of $CO_2$ in the presence of g-$C_3N_4$/$Bi_4O_5I_2$ (20 wt%) with 45.6 $\mu$mol g$^{-1}$ h$^{-1}$ was ~7.9 and ~2.3 times greater than pristine g-$C_3N_4$ and pristine $Bi_4O_5I_2$, respectively. The performance of various BiOX/$Bi_xO_yX_z$-g-$C_3N_4$ heterojunction photocatalysts for the reduction of $CO_2$ is summarized in Table 2.

**Table 2.** Reduction of $CO_2$ in the presence of BiOX/$Bi_xO_yX_z$-g-$C_3N_4$ heterojunction photocatalysts reported in the literature.

| Photocatalyst | Light Source | Result | Significance | Ref. |
|---|---|---|---|---|
| g-$C_3N_4$/$Bi_4O_5I_2$ | 300 W Xe lamp ($\lambda$ > 420 nm) | Photoreduction of $CO_2$ CO—45.6 $\mu$mol g$^{-1}$ h$^{-1}$ | $I_3^-/I^-$ redox mediator assisted Z-scheme mechanism enhanced the photocatalytic $CO_2$ conversion | [184] |
| g-$C_3N_4$/BiOI | 300 W Xe lamp ($\lambda$ > 420 nm) | Photoreduction of $CO_2$, CO—17.9 $\mu$mol g$^{-1}$ h$^{-1}$ $O_2$—9.8 $\mu$mol g$^{-1}$ h$^{-1}$ | Reduction in I content in the composite is unfavourable for the reduction of $CO_2$, implying $I_3^-$ intermediate plays an important role in charge transfer process | [185] |
| g-$C_3N_4$/BiOBr/Au | 300 W Xe lamp ($\lambda$ > 420 nm) | Photoreduction of $CO_2$, CO—6.67 $\mu$mol g$^{-1}$ h$^{-1}$ $CH_4$—0.92 $\mu$mol g$^{-1}$ h$^{-1}$ | The size of Au nanoparticles acted as the Z-scheme bridge and SPR centre during the photocatalytic process. | [139] |
| g-$C_3N_4$/BiOCl-defect rich | 300 W Xe lamp ($\lambda$ > 420 nm) | Photoreduction of $CO_2$, CO—28.4 $\mu$mol g$^{-1}$ h$^{-1}$ $CH_4$—4.6 $\mu$mol g$^{-1}$ h$^{-1}$ | Interfacial oxygen vacancies provide a transport channel for the interfacial carriers, leading to a built-in electric field promoting enhanced carrier transfer efficiency. | [186] |

*4.3. Hydrogen Generation*

Hydrogen as a fuel is considered a promising alternative for future energy sustainability owing to its high specific energy and eco-friendly combustion products. The positive CB position of BiOX photocatalysts restricts their ability to generate $H_2$, but precise control of their thickness during fabrication and the addition of defects such as oxygen vacancies were reported to simultaneously enhance the visible light absorption and intensity of the self-generated IEF [187–190]. For instance, Ye et al. synthesized black coloured ultrathin BiOCl nanosheets enriched with oxygen vacancies while glycerol reacted with the oxygen exposed on the (001) surface under hydrothermal conditions. The amount of $H_2$ generated using the black coloured ultrathin BiOCl (~2.51 $\mu$mol $h^{-1}$) under visible light irradiation was about 21 and 15 times higher than bulk BiOCl (0.12 $\mu$mol $h^{-1}$) and $TiO_2$ (0.16 $\mu$mol $h^{-1}$), respectively [191].

Li et al. reported the growth of BiOCl crystal with 18 facets, 24 vertices and 40 edges through a one-pot hydrothermal method for a longer reaction time of 100 h and, as observed from Figure 7, the amount of photocatalytic $H_2$ generated was 2.1 times greater than that obtained with BiOCl synthesized with a shorter time span of 10 h (5.99 $\mu$mol $g^{-1}$ $h^{-1}$) [192]. Conventionally, the square plates of BiOCl with exposed {001} top facets correspond to the most positive CB position, while the lateral {110} facets form the most negative VB position, facilitating charge separation between in the binary {001}/{110} facet junction. On the other hand, the eighteen-faceted BiOCl were composed of {001} top facets and unusual {102} and {112} oblique facets owing to which the CB position was in the order (001) facet > (102) facet > (112) facet, while the VB position order was (001) facet < (102) facet < (112) facet. Therefore, the well-matched {001}/{102}/{112} ternary facet junction in the eighteen-faceted BiOCl facilitated the efficient cascade charge flow, ensuring enhanced photocatalytic $H_2$ generation. In another report, hierarchical BiOI microspheres synthesized through a microwave-assisted solvothermal method with ethylene glycol and ethanol as solvents were reported to exhibit visible light mediated photocatalytic water splitting to generate $H_2$ with maximum (1316.9 $\mu$mol $g^{-1}$) at pH 7 with a dosage of 0.2 $gL^{-1}$. The narrow bandgap of BiOI (2.04 eV) microspheres, the surprisingly sufficient overpotential due to negative CB position and the higher separation of the photogenerated charges aided $H_2$ generation [193]. Bai et al. synthesized non-stoichiometric $Bi_4O_5X_2$ (X = Br, I) nanosheets through the molecular precursor method, which generated $H_2$ under a 300 W lamp emitting simulated solar light irradiation [67]. Using 10% methanol as the sacrificial agent, the amount of $H_2$ generated with 40 mg of $Bi_4O_5Br_2$ and $Bi_4O_5I_2$ was 4.21 and 2.79 $\mu$mol $g^{-1}$ $h^{-1}$, respectively. The enhanced photocatalytic performance of $Bi_4O_5Br_2$ nanosheets with a quantum efficiency of 0.93% at 420 nm in comparison to $Bi_4O_5I_2$ (just 0.52%) was attributed to the greater separation of photogenerated charge carriers. $Bi_{24}O_{31}Br_{10}$ nanoplates synthesized through the chemical precipitation method generated $H_2$ by the photocatalytic reduction of water at a rate of 3.3 $\mu$mol $h^{-1}$ with 50 mg catalyst loading, while pristine BiOBr and $Bi_2O_3$ displayed no activity. The uplifting of the CB of $Bi_{24}O_{31}Br_{10}$ due to the presence of Bi 6p and Br 4s orbitals fulfilled the electric potential requirements for splitting water to $H_2$ in comparison to pristine BiOBr and $Bi_2O_3$ with positive CB positions [68]. Di et al. reported the synthesis of a defect-rich single-unit-cell of $Bi_3O_4Br$ with a thickness of ~1.7 nm that displayed superior photocatalytic $H_2$ generation of up to 380 $\mu$mol $g^{-1}$ $h^{-1}$, which was ~2 and 4.9 times greater than defect-deficient $Bi_3O_4Br$ and bulk $Bi_3O_4Br$, respectively [97]. The enhanced photocatalytic activity of defect-rich single-unit-cell $Bi_3O_4Br$ was immensely facilitated by the generation of oxygen defects due to bismuth vacancy in addition to their atomically thin architecture that favourably tuned the electronic band structure. In another study, a bilayer junction formed by the selectively assembly of metallic phase enriched $MoS_2$ and oxygen-deficient $Bi_{12}O_{17}Cl_2$ monolayers exhibited photocatalytic $H_2$ evolution at a rate of 33 mmol $h^{-1}$ $g^{-1}$ under visible light, with a superior quantum efficiency of 36% at 420 nm that was superior to the pristine monolayers of $MoS_2$ and $Bi_xO_yX_z$-based systems [194]. The enhanced performance of the bilayer $MoS_2$/$Bi_{12}O_{17}Cl_2$ junction can be attributed to the enhanced charge separation

in oxygen-deficient $Bi_{12}O_{17}Cl_2$ monolayers ensured by the IEF and the collective role of both IEF and Bi-S bonds for pushing the electrons to catalyse the $H_2$ evolution. However, surprisingly, to date no reports have been found on photocatalytic $H_2$ generation using a $BiOX/Bi_xO_yX_z$-g-$C_3N_4$ heterojunction photocatalyst, given the fact that pristine g-$C_3N_4$ is an excellent $H_2$ evolution photocatalyst that has been studied extensively.

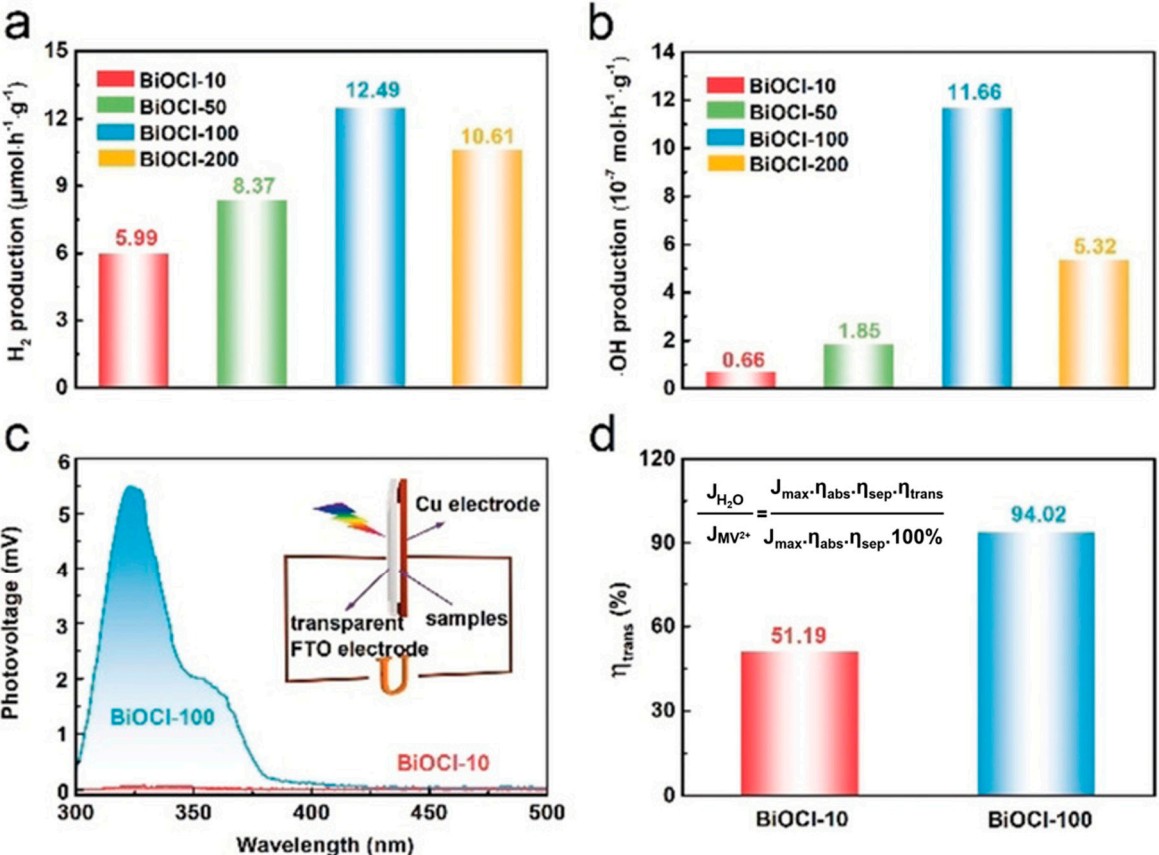

**Figure 7.** Rate constants for generation of (**a**) $H_2$ and (**b**) hydroxyl ions over different BiOCl photocatalysts under simulated solar light irradiation. (**c**) Surface photovoltage spectra of BiOCl synthesized with time span of 10 h (BiOCl-10) and 100 h (BiOCl-100) indicating the degree of charge separation. (**d**) Surface charge transfer efficiency of BiOCl-10 and BiOCl-100. Reprinted from Ref. [192] with permission from John Wiley & Sons.

*4.4. Oxygen Evolution*

In addition to $H_2$ generation, sunlight-driven photocatalytic water splitting allows the generation of oxygen ($O_2$). However, the water oxidation for $O_2$ evolution is more difficult due to the multistep transfer of four $h^+$ in comparison to the transfer of two $e^-$ for $H_2$ generation. The oxygen evolution reaction demands the accumulation of cationic $h^+$ on the surface (i.e., surface-trapped holes), which is absolutely essential to be utilized for the reduction of adsorbed water via $H_2O_{ad} + 2h^+ \rightarrow {}^1/_2\,O_2 + 2H^+$ [195]. The basic requirement for photocatalytic $O_2$ evolution is ensuring that the VB edge of the photocatalyst is located at a more positive position than the oxidation potential of $H_2O$ (1.23 V vs. normal hydrogen electrode at pH = 0). Further, a significant overpotential is required for overcoming the activation energies in the charge-transfer process between the photocatalyst and water molecules. Due to the stringent demands, only very few materials are capable of directly oxidizing water into $O_2$ under light irradiation. Di et al. reported the fabrication of atomically thin defect-rich BiOCl nanosheets through the hydrothermal approach by treating pre-synthesized BiOCl nanosheets in ethylene glycol and studied their performance towards the photooxidation of $H_2O$ [196]. The amount of $O_2$ generated with defect-rich BiOCl nanosheets (56.85 $\mu$mol g$^{-1}$ h$^{-1}$) was nearly 3 and 8 times greater than that gen-

erated with defect-free BiOCl nanosheets and bulk BiOCl. The enhanced performance of defect-rich BiOCl in the photooxidation of water can be attributed to the synergetic effect of an atomically thin thickness of ~2 nm, defects on BiOCl basal planes shortening the migration distance of holes for promoting charge separation and hole utilization, and the presence of abundant coordination-unsaturated active atoms. In another study, Ag and $PdO_x$ nanocubes selectively deposited on the (001) and (110) facets of BiOCl nanoplates formed a ternary hybrid Ag-BiOCl-$PdO_x$ photocatalyst that was employed in the photocatalytic $O_2$ evolution under visible light with $NaIO_3$ as the electron sacrificial agent [197]. Interestingly, Ag-(110)BiOCl(110)-$PdO_x$ exhibited a highest average $O_2$ rate of 68.2 µmol $g^{-1}$ $h^{-1}$, which was almost 5.9, 1.9 and 1.6 times higher than Ag-(001)BiOCl(001)-$PdO_x$, Ag-(001)BiOCl(110)-$PdO_x$ and Ag-(110)BiOCl(001)-$PdO_x$, respectively. The schematic in Figure 8 illustrates the reasons for the enhanced photocatalytic $O_2$ generation, which can be attributed to stronger electronic coupling at the BiOCl(110)-based interfaces as a result of the thinner contact barrier between Ag and $PdO_x$ and the shortest average hole diffusion distance realized by Ag and $PdO_x$ on the BiOCl(110) plane.

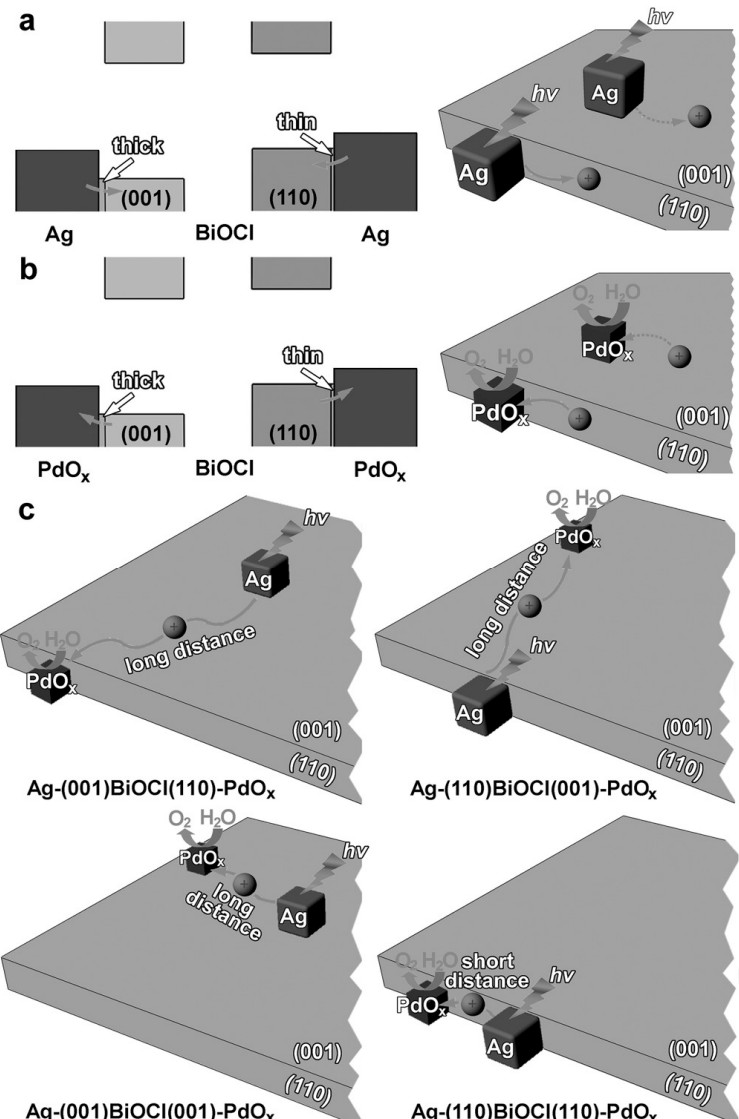

**Figure 8.** Schematic depicting the facet-dependent interfacial hole transfer ability as a result of the difference in thickness of the contact barrier layer on (**a**) Ag-BiOCl and (**b**) BiOCl-$PdO_x$ interfaces. (**c**) Schematic representation of the different average diffusion distances of hole in different Ag-BiOCl-$PdO_x$ photocatalysts. Reprinted from Ref. [197] with permission from John Wiley & Sons.

Cui et al. reported the solvothermal synthesis of BiOCl nanosheets with abundant oxygen vacancies using ethylene glycol as the solvent and studied their photooxidation ability [198]. The rate of $O_2$ evolved under visible light with oxygen vacancy-rich BiOCl nanosheets in the presence of $AgNO_3$ as the electron acceptor was 1.72 mmol $g^{-1}$ after 5 h, which was 3.3 times higher than oxygen vacancy-poor BiOCl nanosheets despite the fact that their surface area was almost identical. Abundant oxygen vacancies in BiOCl nanosheets were reported to create many electron donor levels and allowed the excitation of electrons, which subsequently formed holes in the VB for the $O_2$ evolution reaction. Similarly, Ji et al. reported the synthesis of oxygen vacancy-rich and oxygen vacancy-less $Bi_7O_9I_3$ microspheres through the ionic liquid assisted solvothermal method and studied their performance in the photocatalytic $O_2$ evolution. As expected, the $O_2$ evolution rate of oxygen vacancy-rich $Bi_7O_9I_3$ microspheres at 199.2 μmol $g^{-1}$ $h^{-1}$ was ~1.5 times greater than that of oxygen vacancy-less $Bi_7O_9I_3$ microspheres, despite the fact that their surface area was comparable [199]. In another study, $Bi_3O_4Br$ nanorings were synthesized through the solvothermal method using cetyltrimethylammonium bromide and polyvinyl pyrrolidone as surfactants, and their performance was assessed through photocatalytic $O_2$ evolution.

Interestingly, $Bi_3O_4Br$ exhibited $O_2$ efficient oxygen evolution at a rate of 72.54 μmol $g^{-1}$ $h^{-1}$ that was attributed primarily to its single-crystalline nature, (001) facets exposure, ring structure, appropriate light response range and band potential, which facilitated the migration of charge carriers [200]. Ning et al. constructed a 2D-2D heterostructure photocatalyst by coupling $Bi_3O_4Cl$ and BiOCl nanosheets through alkaline chemical etching and solvent exfoliation for $O_2$ evolution under visible light [201]. The rate of $O_2$ evolved with ultrathin $Bi_3O_4Cl$/BiOCl in the presence of $FeCl_3$ as the electron scavenger reached 58.6 μmol $g^{-1}$ $h^{-1}$, which was about 3 times higher than that of nanocrystal $Bi_3O_4Cl$/BiOCl. Electron spin resonance spectroscopy detected $^{\bullet}O_2^{-}$ as the primary active species, which strongly suggested the mechanism of charge transfer during the photocatalytic oxidation reaction to be the Z-scheme heterojunction. In the $Bi_3O_4Cl$/BiOCl Z-scheme heterojunction, photogenerated electron-hole pairs generated by the built-in electric field under visible light irradiation enabled the rapid transfer of photogenerated electrons to the {001}-BiOCl facets that were partly trapped by $Fe^{3+}$, while the holes gathered on the {001}-$Bi_3O_4Cl$ facets accommodated plenty of active sites for the photocatalytic $O_2$ evolution [201]. Though g-$C_3N_4$ has been extensively studied for its ability to oxidize water under light irradiation [202–204], it is unfortunate that no work on photocatalytic water oxidation has been carried out by designing suitable BiOX/$Bi_xO_yX_z$-g-$C_3N_4$ heterojunction photocatalysts. However, there is enough scope for constructing efficient heterojunction photocatalysts using BiOX/$Bi_xO_yX_z$ with exposed facets and functionalized g-$C_3N_4$ that could achieve enhanced quantum efficiencies.

*4.5. Nitrogen Reduction*

The photoreduction of nitrogen ($N_2$) to produce ammonia ($NH_3$), commonly referred to as nitrogen fixation, is a green alternative to the standard Haber–Bosch process, which consumes large amounts of fossil fuels and releases $CO_2$ into the atmosphere. Li et al. reported the solvothermal synthesis of {001} facet exposed BiOBr nanosheets with and without oxygen vacancies for studying their photocatalytic performance in reducing $N_2$ under visible light irradiation with water as the solvent and proton source. Interestingly, {001}-BiOBr without oxygen vacancies did not exhibit photocatalytic activity, while {001}-BiOBr with oxygen vacancies generated a significant amount of $NH_3$ at rate of 104.2 and 223.3 μmol $g^{-1}$ $h^{-1}$ under visible light and UV-vis light irradiation, respectively, with an external quantum efficiency of 0.23% at 420 nm [205]. $N_2$ was adsorbed on the oxygen vacancies by combining with the two nearest Bi atoms in the sublayer to form a terminal end-on bound structure, and the reduction capacity of $N_2$ over {001}-BiOBr was directly dependent on the amount of oxygen vacancies as they acted as catalytic centres capable of adsorbing and activating $N_2$ by inhibiting electron-hole recombination

and promoting the interfacial charge transfer. Similarly, the Zhang group also studied photocatalytic $N_2$ fixation using oxygen vacancy-rich BiOCl nanosheets with {001} and {010} exposed facets and, interestingly, it was found that the rate of $NH_3$ generation with {010}-BiOCl (0.95 $\mu mol\ g^{-1}\ h^{-1}$) was only half of {001}-BiOCl (1.89 $\mu mol\ g^{-1}\ h^{-1}$), but after 30 min the rate of $NH_3$ generation with {010}-BiOCl at 2.29 $\mu mol\ g^{-1}\ h^{-1}$ was 1.21 times greater than {001}-BiOCl. The reason for the slower rate of $NH_3$ generation during the initial 30 min was attributed to the different chemistry of $N_2$ fixation on {001} and {010} facets, while the enhanced $NH_3$ generation was attributed to the more stable side-on bridging of $N_2$ by combining with the two nearest Bi atoms in the outer layer and the nearest Bi atom in the sublayer on the (010) surface [206]. Bai et al. synthesized bismuth-rich $Bi_5O_7I$ with {001} and {100} exposed facets through the solvothermal treatment of molecular precursors in glycerol, and studied their photocatalytic activity for $N_2$ fixation. The $NH_3$ generation rate using {001}-$Bi_5O_7I$ (111.5 $\mu mol\ g^{-1}\ h^{-1}$) was ~2.3 times greater than {100}-$Bi_5O_7I$ (47.6 $\mu mol\ g^{-1}\ h^{-1}$), and the apparent quantum efficiency was 5.1% at 365 nm. Band structure studies through VB X-ray photoelectron spectroscopy revealed the more negative CB position of {001}-$Bi_5O_7I$ nanosheets that enhanced their reduction power, while the photocurrent response and electrochemical impedance spectroscopy results indicated their enhanced separation of photogenerated charge carriers and lower resistance for electron-transfer. Therefore, it was concluded that the enhanced photocatalytic $N_2$ fixation in Bi-rich $Bi_xO_yX_z$ was due to the facet effect in comparison to BiOX, wherein oxygen vacancies play a dominant role [207]. Another study on Bi-rich $Bi_xO_yX_z$ reported by Wang et al. demonstrated that engineering oxygen vacancies into $Bi_5O_7Br$ nanotubes with a uniform diameter of ~5 nm could generate $NH_3$ up to 1.38 mmol $h^{-1}$ $g^{-1}$ under visible light with pure water without any organic scavengers or cocatalysts with an apparent quantum efficiency of over 2.3% at 420 nm [83]. Interestingly, the $Bi_5O_7Br$ nanotube dispersion in water exhibited a colour change from light yellow to dark grey under light irradiation that induced oxygen vacancies by seizing O atoms from water. In addition to the more negative CB position, the enhanced chemisorption of $N_2$ on the oxygen vacancy sites due to the large surface area of $Bi_5O_7Br$ nanotubes (96.56 $m^2\ g^{-1}$), forming a bond with Bi-metal (sideward transition metal), enabled it to donate electrons from its bonding orbitals and accept electrons to its antibonding $\pi$-orbitals, which gradually wakened the N-N triple bond due to electron exchange and led to the enhanced generation of $NH_3$. Zhang et al. reported photocatalytic $N_2$ fixation by simultaneously introducing oxygen vacancy and doping Fe into BiOCl nanosheets that generated $NH_3$ at a rate of 1.02 mmol $g^{-1}\ h^{-1}$ under light irradiation using a 300 W Xe lamp [208]. The mechanism of $N_2$ fixation was similar to the report on $Bi_5O_7Br$ nanotubes, and the dispersion of Fe-doped BiOCl also exhibited a colour change from white to dark grey under light irradiation for the generation of oxygen vacancies. Typically, the $N_2$ fixation involved four main steps, viz., (i) the generation of oxygen vacancies on the catalyst surface during light irradiation, (ii) the chemisorption of $N_2$ on the catalyst surface activated by oxygen vacancies, (iii) the injection of photogenerated electrons into the orbitals of activated $N_2$ for their reduction, and (iv) the refilling of oxygen vacancies by adjacent O atoms from $H_2O$ or $O_2$. Similarly, Fe-doped BiOBr microspheres composed of nanosheets were synthesized through the solvothermal method with polyethylene glycol for the photocatalytic conversion of $N_2$ to $NH_3$ at a rate of 382.68 $\mu mol^{-1}\ g^{-1}\ h^{-1}$, which was eight times greater than pristine BiOBr (51.6 $\mu mol^{-1}\ g^{-1}\ h^{-1}$), under visible light radiation obtained from a 300 W xenon lamp equipped with a 420 nm cutoff filter [209]. The charge density map of Fe-doped BiOBr nanosheets shown in Figure 9a indicated that Fe withdrew electrons from nearby atoms to form electron-rich Fe(II) that injected localized electrons to the $\pi$ N-N antibonding orbital of the adsorbed $N_2$ via electron donation for obtaining enhanced $NH_3$ generation, as observed from Figure 9b. Further, the more negative CB position of Fe-doped BiOBr nanosheets (Figure 9c) in comparison to pristine BiOBr and enhanced visible light absorption demonstrated the vital role played by Fe atoms. Similar to defect-rich nanostructures of BiOX and $Bi_xO_yX_z$, defect-rich $g$-$C_3N_4$ has demonstrated excellent performance in the

photocatalytic $N_2$ fixation under visible light [210,211]. However, to date no work has been reported on photocatalytic $N_2$ fixation with suitable $BiOX/Bi_xO_yX_z$-g-$C_3N_4$ heterojunction photocatalysts, which allows room for significant research to be conducted in this direction.

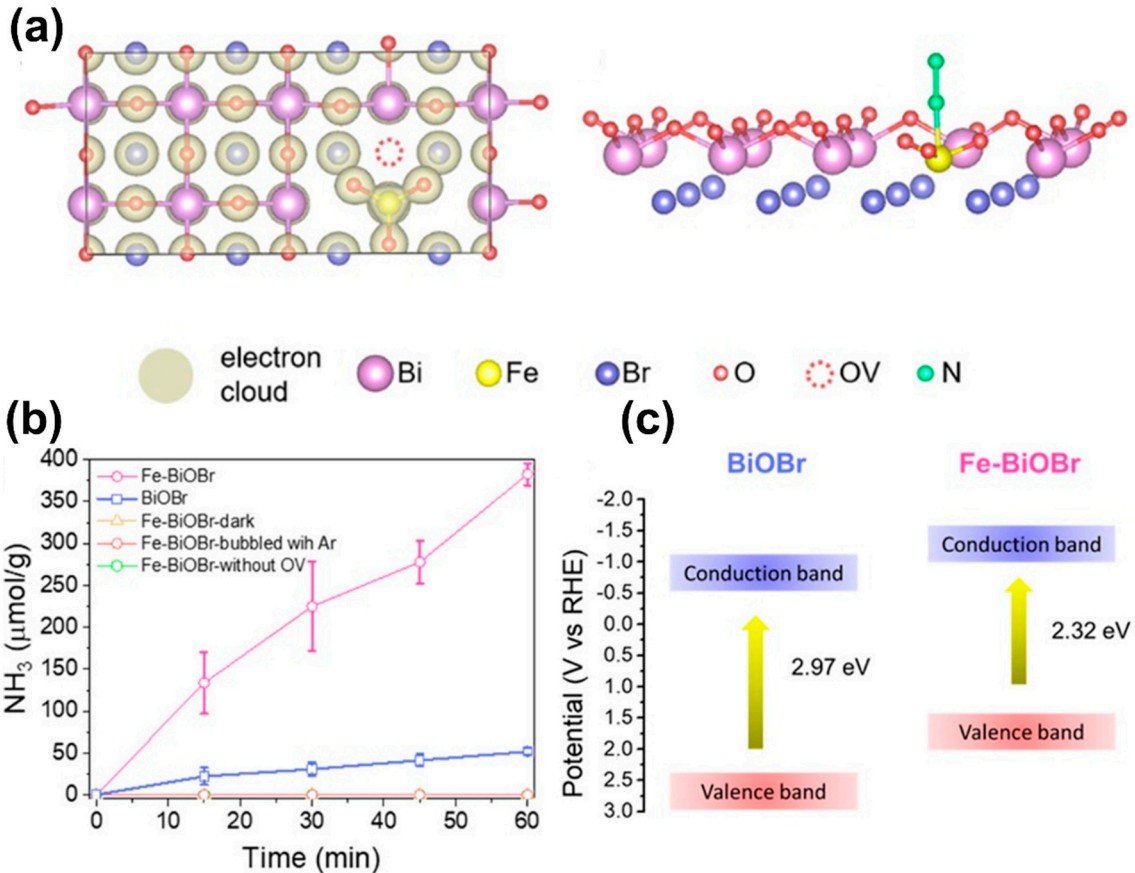

**Figure 9.** (**a**) Charge density map of Fe-doped BiOBr nanosheets and the schematic of $N_2$ binding to the oxygen vacancy connected Fe atom in Fe-BiOBr, (**b**) plot depicting the photocatalytic $NH_3$ generation using Fe-doped BiOBr nanosheets and (**c**) band structure of BiOBr and Fe-doped BiOBr nanosheets. Reprinted from Ref. [209] with permission from American Chemical Society.

### 4.6. Organic Synthesis

Semiconductor-based photocatalysis to achieve highly efficient organic reaction has gained significant research attention. The oxidation of alcohols to their corresponding aldehydes is the major area of organic synthesis. Xiao et al. hydrothermally synthesized nanobelt-like structures of $Bi_{12}O_{17}Cl_2$ and evaluated their performance towards the photocatalytic oxidation of benzyl alcohol in acetonitrile to benzaldehyde under visible light at 50 °C. The bandgap energy of the $Bi_{12}O_{17}Cl_2$ photocatalyst was found to be 2.43 eV, and the conversion rate to benzaldehyde was 44% under oxygen atmosphere via direct hole oxidation. When $Bi_{12}O_{17}Cl_2$ nanobelts were subjected to visible light irradiation, the $e^-$ excited to the CB would be trapped by electrophilic $O_2$, while the $h^+$ in the VB reacted with alkoxide anions to form carbon radicals through deprotonation, and, subsequently, benzaldehyde was formed by the reaction of these carbon radicals with $h^+$ [212]. Han et al. reported the synthesis of BiOBr photocatalysts with three different exposed facets, viz., {001}, {010} and {110} for the selective aerobic photooxidation of benzylamine in acetonitrile solution to N-benzylidenebenzylamine at room temperature and atmospheric air as the oxidizing agent. Although BiOCl and BiOI were found to exhibit almost 100% selectivity for the photooxidation of benzylamine, only BiOBr exhibited 100% conversion and selectivity. Results indicated that the orientation of the exposed planes played a significant role as BiOBr-{001} exhibited the highest activity based on unit surface area. However,

solvothermally synthesized BiOBr-{110} microspheres achieved 100% selectivity and conversion efficiency in the oxidation of benzylamine due to their high surface area [213]. BiOBr nanoplates with (001) exposed facets synthesized through the modified hydrothermal approach were treated in $O_2$ and inert atmosphere for fabricating defect-free and defect-rich BiOBr. Interestingly, the defect-rich BiOBr nanoplates exhibited high efficiency and selectivity for the oxidation of benzylamine to N-benzylidenebenzylamine, while the yield of corresponding imine was much lower with defect-free BiOBr. Photoluminescence spectroscopy and photoelectrochemical studies confirmed that oxygen-vacancy mediated exciton dissociation resulted in promoted charge-carrier generation in the system that led to a selective oxidative-coupling reaction through $^\bullet O_2{}^-$ generation [214]. Similarly, BiOCl colloidal ultrathin nanosheets with hydrophobic surface properties fabricated with abundant oxygen vacancies by the hydrolysis of $BiCl_3$ in octadecylene solution enabled them to display superior photocatalytic activity for the aerobic oxidation of secondary amines to corresponding imines under visible light irradiation [215]. $Bi_{24}O_{31}Br_{10}(OH)_\delta$ microspheres containing porous nanosheet substructures with a surface area of $45 \text{ m}^2\text{g}^{-1}$ and abundant active lattice oxygen sites were reported to exhibit the selective photooxidation of various alcohols in air under visible light irradiation [216]. Mott–Schottky analysis suggested the thermodynamically feasible band structure of $Bi_{24}O_{31}Br_{10}(OH)_\delta$, while its loose and porous architecture allowed the easy diffusion of bulky alcohols for accessing the abundant active surface sites. Therefore, a remarkably high quantum efficiency of 71% was achieved under visible light irradiation for isopropanol oxidation. A BiOBr/g-$C_3N_4$ heterojunction photocatalyst synthesized through a two-step combustion-coprecipitation method was reported to exhibit excellent photooxidation of benzylamine to N-benzylidenebenzylamine with a conversion rate of 94% and a yield of 82% within 4 h of visible light irradiation obtained from white LED under atmospheric air [217]. The enhanced performance of the BiOBr/g-$C_3N_4$ photocatalyst was ascribed to the improved charge transfer and separation driven by its apt band structure. Interestingly, bezylamine oxidation happened under both aerobic and anaerobic conditions driven by the $^\bullet O_2{}^-$ radicals (produced by the reaction of CB $e^-$) with amine cations and the reaction of VB $h^+$ with nitrogen-centred radicals, respectively, to form N- benzylidenebenzylamine.

## 5. Strategies for Improving the Performance of BiOX/$Bi_xO_yX_z$-g-$C_3N_4$ Heterojunction Photocatalysts

The photocatalytic performance of BiOX/$Bi_xO_yX_z$ nanomaterials has received substantial research interest owing to their suitable band structure for absorbing sunlight to start the photocatalytic reaction. Unfortunately, their practical applications are still confined by a few drawbacks, including a mismatch between the band edge position and light harvesting, ineffective charge separation and transportation, fewer active sites, and poor selectivity of the desired reaction. In this context, numerous strategies have been developed to engineer the layered structure and overcome the aforementioned drawbacks. The following sections emphasize each strategy accordingly.

### 5.1. Microstructure Modulation

Due to the strong connection between the physical and chemical properties and the microstructure (shape, size, surface area, and dimensionality) of the materials, the rational synthesis of the nano- or microstructure has constantly received great significance from the prospect of both scientific research and industrial applications. Further, the inherent nature of nanoscale materials to exhibit higher surface-to-volume ratio and provide abundant active sites enables the effective separation of the photoinduced carriers, thereby enhancing their photocatalytic efficiency. Table 1 provides the summary of various methods for fabricating BiOX/$Bi_xO_yX_z$-g-$C_3N_4$ heterojunction photocatalysts which were briefly introduced in Section 4.1. Since many articles have already reviewed the importance of microstructure modulation, our discussions in this section are confined to just a few articles mainly focusing on the fabrication of the heterojunction between BiOX/$Bi_xO_yX_z$

and g-$C_3N_4$. In addition to the conjunction 2D-2D heterojunction, the embedment of 3D hierarchical structures on 2D structures has also sparked interest owing to the distinctive 3D architecture formed by the self-assembly of 1D and 2D sub-structures.

For example, a $Bi_5O_7I$/g-$C_3N_4$ heterojunction photocatalyst was synthesized by two different approaches, adopting in situ co-thermolysis [106] and the one-pot ethylene glycol assisted hydrothermal approach [107]. In the in situ co-thermolysis method, BiOI precursor (pre-synthesized through the coprecipitation method) was mixed with melamine and ground with an agate mortar, and the powdered material taken in a crucible was heated in a muffle furnace at 520 °C for 4 h for obtaining $Bi_5O_7I$/g-$C_3N_4$. On the other hand, in the one-pot ethylene glycol assisted hydrothermal approach, a final solution of ethylene glycol made by the dropwise addition of KI solution to a solution containing $Bi(NO_3)_3 \bullet 5H_2O$ with pre-synthesized g-$C_3N_4$ was treated hydrothermally at 150 °C for 12 h. Interestingly, the morphology of the final structure of $Bi_5O_7I$/g-$C_3N_4$ resembled microspheres with nanosheet substructures of the individual components. Additionally, interestingly, the estimated values of the VB and CB potentials for g-$C_3N_4$ (1.54 eV and $-1.19$ eV) and $Bi_5O_7I$ (3.17 eV and 0.29 eV) were identical. However, the mechanism of charge transfer described for the $Bi_5O_7I$/g-$C_3N_4$ heterojunction photocatalyst synthesized through in situ co-theromolysis was ascribed to the type-II heterojunction, while the charge transfer mechanism in hydrothermally synthesized $Bi_5O_7I$/g-$C_3N_4$ was ascribed to the Z-scheme. In another study, BiOBr/g-$C_3N_4$ heterojunction photocatalysts were fabricated by dispersing pre-synthesized BiOBr nanoflowers enriched with oxygen vacancies synthesized by the solvothermal treatment of precursors ($Bi(NO_3)_3 \bullet 5H_2O$, polyvinylpyrrolidone and KBr) dispersed in mixed solvent (ethylene glycol and water) at 160 °C for 3 h, in g-$C_3N_4$ dispersion and stirring at room temperature for 6 h, followed by washing and drying [218].

TEM analysis indicates the layered g-$C_3N_4$ structure with ultrathin nanosheets (Figure 10a), the nanoflower-like morphology of BiOBr enriched with oxygen vacancies (Figure 10c) and their perfect heterojunction, indicating the embedment of the oxygen vacancy enriched nanoflowers on g-$C_3N_4$ nanosheets (Figure 10d). Comparatively, the morphology of defect-free BiOBr/g-$C_3N_4$ indicates the formation of nanoplates, as observed from Figure 10b. HRTEM micrographs (Figure 10e,f) of oxygen vacancy enriched BiOBr/g-$C_3N_4$ depict the lattice spacing of d = 0.28 and 0.352 nm corresponding to the (102) and (101) crystal planes of the tetragonal phase of BiOBr, respectively. Further, the purity and co-existence of all the elements in BiOBr/g-$C_3N_4$ were confirmed from EDS elemental mapping, as shown in Figure 10g. The photocatalytic activity of oxygen vacancy enriched BiOBr/g-$C_3N_4$ in the removal of NO under visible light irradiation at 63% was 1.8, 1.6, 1.6 and 1.5 times greater than pristine g-$C_3N_4$, pristine oxygen vacancy enriched BiOBr, defect-free BiOBr/g-$C_3N_4$ and a physical mixture of g-$C_3N_4$ with oxygen vacancy enriched BiOBr. Similarly, photocatalytic $CO_2$ reduction using oxygen vacancy enriched BiOBr/g-$C_3N_4$ generated CO and $CH_4$ at a rate of 61.8 and 27.1 $\mu molh^{-1}g^{-1}$, respectively, which was greater than the control samples. Abundant oxygen vacancies in BiOBr and the heterojunction with ultrathin g-$C_3N_4$ nanosheets were attributed to the enhanced photocatalytic activity, while the $^\bullet OH$ and $^\bullet O_2{}^-$ radicals were reported to be the main active species involved in the removal of NO and the reduction of $CO_2$, respectively.

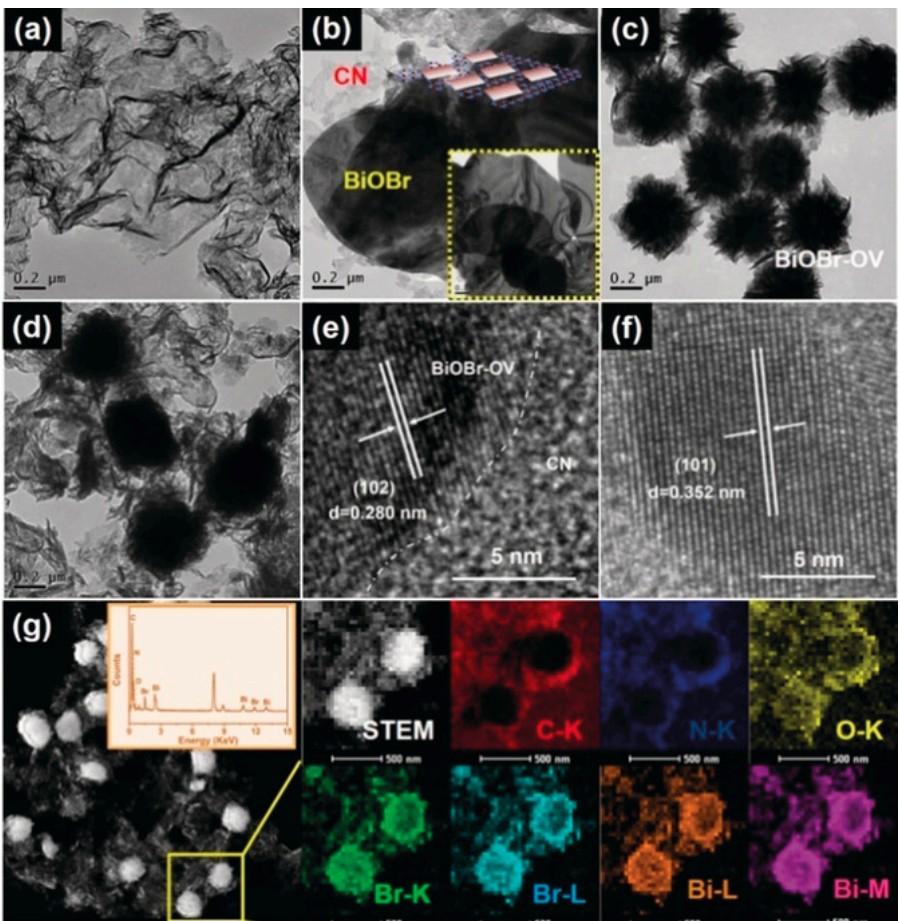

**Figure 10.** TEM micrographs of (**a**) pristine g-C$_3$N$_4$, (**b**) defect free-BiOBr/g-C$_3$N$_4$, (**c**) pristine BiOBr enriched with oxygen vacancies, (**d**) oxygen vacancy enriched BiOBr/g-C$_3$N$_4$, (**e**,**f**) HRTEM micrographs of oxygen vacancy enriched BiOBr/g-C$_3$N$_4$ depicting the lattice spacing and (**g**) the corresponding elemental maps of C, N, O, Br and Bi. Reprinted from Ref. [218] with permission from Wiley-VCH.

### 5.2. Facet and Defect Control

Crystal facets are an important feature of crystalline materials, and different crystal facets have different geometric and electronic structures, exhibiting intrinsic reactivity and surface physical and chemical properties associated with the crystallographic orientation. As a basic feature of crystalline materials, the exposed crystal facets play an important role in photocatalytic efficiency since photocatalysis occurs on the surface of BiOX photocatalysts. BiOCl nanosheets with tunable {001} facet percentages were synthesized by hydrolyzing molecular precursors Bi$_n$(Tu)$_x$Cl$_{3n}$ (Tu = thiourea). Exposed {001} facets of BiOCl exhibited high oxygen atom density, and under UV light irradiation, plenty of oxygen vacancy sites were created [219]. These oxygen vacancies formed a defect state near the bottom of the CB of BiOCl and played a significant role in capturing the photogenerated electrons for enhancing the photocatalytic activity of BiOCl due to the improved separation of photogenerated charge carriers. Jiang et al. reported the hydrothermal synthesis of BiOCl single-crystalline nanosheets with exposed {001} facets, which exhibited higher activity for direct semiconductor photoexcitation pollutant degradation under UV light, while the counterpart with exposed {010} facets possessed superior activity for indirect dye photosensitization degradation under visible light [220]. Zhao et al. obtained rose-like BiOBr nanostructures with exposed {111} facets using sodium dodecyl sulphate as the surfactant, which exhibited better photocatalytic activity than exposed {001} facets under both visible light and monochromatic light [221]. Although high-energy facets exhibited higher

activity than low-energy facets, they are easily eliminated because the fastest crystal growth would occur in the direction perpendicular to the high-energy facet. Therefore, glucose as the capping and structure-directing agent was employed in synthesizing 1D rod-like BiOBr with exposed {110} facets, and it was revealed that glucose not only suppressed the growth of {001} facets of BiOBr nanosheets but also induced these nanosheets to self-assemble along the [1] orientation, displaying better photocatalytic activity towards the photodegradation of rhodamine B and methyl orange [222]. Defects in the exposed facets of semiconductors can significantly enhance the photocatalytic activity by changing their electronic structures, the recombination efficiency of charge carriers, and surface properties [223]. As a typical defect, oxygen vacancies are reported to enhance the photo-absorption and photocatalytic performance of the photocatalysts. Li et al. reported the fabrication of BiOBr nanosheets with oxygen vacancies via a hydrothermal-reduction route. Their study revealed that only those oxygen vacancies created on the surface of the photocatalyst could inhibit the charge carrier recombination by trapping the photogenerated electrons, while the bulk oxygen vacancies which can also trap photogenerated charges act as recombination centres, resulting in a decrease in photoactivity [205]. Wang et al. reported the introduction of surface oxygen vacancies over the BiOBr nanosheets exposed with {001} facets by surface modification using polybasic carboxylic acids. These surface oxygen vacancies on BiOBr intensified the separation efficiency of photogenerated carriers and promoted the dioxygen reduction towards the degradation of MO dye [224]. Further, density functional theory calculations revealed that the presence of oxygen vacancies can ensure the increased density of states at the conduction band edge relative to the BiOBr atomic layers and bulk counterpart, which helps in enhancing the electron transport pathways.

Additionally, the introduced oxygen vacancies created new defect levels which allowed a narrower bandgap, hence giving the possibility for realizing visible light $CO_2$ reduction. Wu et al. reported that oxygen-deficient BiOBr atomic layers triggered visible-light-driven $CO_2$ reduction into CO with a rate of 87.4 $\mu$mol g$^{-1}$ h$^{-1}$, which was 20 times and 24 times higher than that of BiOBr atomic layers and bulk BiOBr. Thus, defect engineering was proved to promote $CO_2$ photoreduction efficiency through fully addressing the poor photo-absorption, sluggish electron-hole separation, and high $CO_2$ activation barrier, giving new possibilities for achieving high performance in solar $CO_2$ reduction [225].

Li et al. reported the synthesis of a BiOCl single crystal with eighteen-facets by prolonging the hydrothermal reaction time (10–200 h), which exhibited enhanced $H_2$ generation that was higher than previously reported BiOCl with a [1] top facet and [121] lateral facets. SEM micrographs of BiOCl crystals synthesized at different time intervals are represented in Figure 11a–d. Both TEM and HRTEM micrographs confirmed the formation of well-shaped oblique facets at an angle of ~45° from the top facets, as observed from Figure 11e–h. The schematic illustration of eighteen-faceted BiOCl and the {001}, {102} and {112} facets in BiOCl is represented in Figure 11i,j. The well-indexed XRD patterns indicated the formation of a pure phase of BiOCl (Figure 11k). Therefore, with the help of the ternary facet junction, the electron-hole pairs in the eighteen-faceted BiOCl single crystal were effectively separated and displayed outstanding photocatalytic activity in the generation of $H_2$ [192].

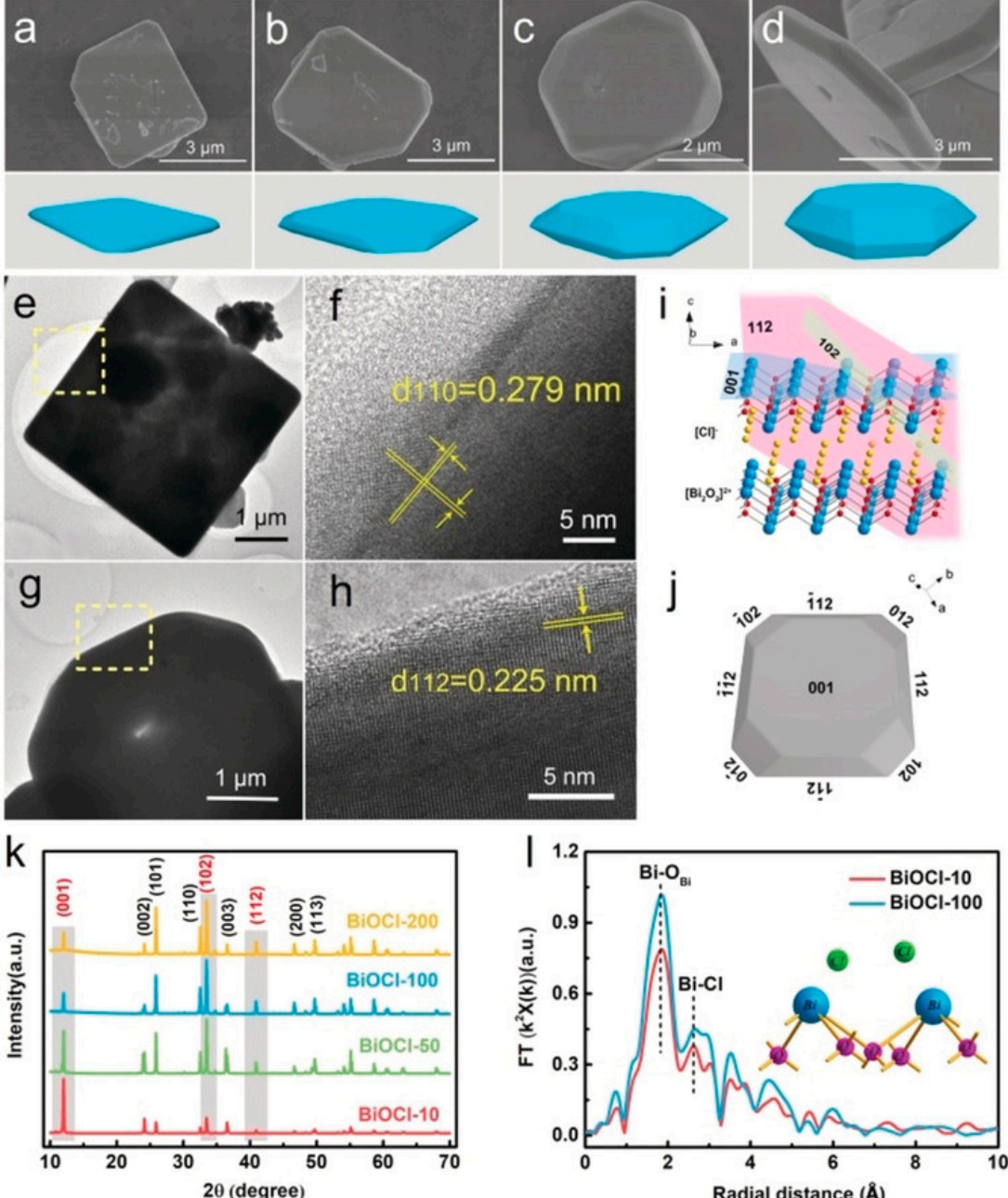

**Figure 11.** SEM images and the corresponding schematic representation for BiOCl treated at different time intervals (**a**) 10 h, (**b**) 50 h, (**c**) 100 h, and (**d**) 200 h; (**e**) TEM image and (**f**) HRTEM image of BiOCl treated for 10 h; (**g**) spherical aberration correction TEM image and (**h**) HRTEM image of BiOCl treated for 100 h; (**i**) crystal structure and facets of BiOCl; (**j**) Schematic representation of the different facets of eighteen-faceted BiOCl; (**k**) XRD patterns of BiOCl series samples; (**l**) Fourier transformed profiles for Bi coordination environments in normalized Bi L3-edge XAFS spectra of BiOCl treated at 10 and 100 h. Reprinted from Ref. [192] with permission from John Wiley & Sons.

### 5.3. Integration with Noble Metal Nanostructures

Depositing the noble metal over the surface of the semiconductor surface is an effective approach for modifying the photon harvesting capacity and for increasing the charge carrier separation kinetics. Noble metals coupled with semiconductor photocatalysts could form a high-speed charge-transfer channel for accelerated transport. Further, in many semiconductor systems, the noble metal nanoparticles have been usually used as charge-transfer mediators owing to their excellent electron conductivity, thereby offering a new approach to overcome the limit of traditional heterojunction photocatalysts. Additionally, due to the

unique phenomenon of surface plasmon resonance (SPR) and its induced local electric field, the noble metal nanoparticles bridged with semiconductor photocatalysts can strengthen the photon absorption range and boost the photoinduced electron transfer [226,227]. Jiang et al. coupled Pt nanoparticles as the apt co-catalyst in the ternary Z-scheme photocatalytic system with BiOI (5.65 eV) and g-$C_3N_4$ (4.52 eV), since the work function of Pt (5.20 eV) was in between that of the individual semiconductors. Interestingly, the efficiency of the BiOI/Pt/g-$C_3N_4$ system was much higher than that of pristine g-$C_3N_4$, Pt/g-$C_3N_4$, and BiOI/g-$C_3N_4$ in the photodegradation of phenol and tetracycline hydrochloride, which was attributed to the efficient separation and transfer of charge carriers in an unobstructed Z-scheme route. The electric fields in the opposite direction for Pt/BiOI and Pt/g-$C_3N_4$ interfaces were formed due to the difference in the work function. The higher work function of Pt in contact with g-$C_3N_4$ formed a Schottky barrier and forced the transfer of electrons accumulated in the space charge region, resulting in the upward band bending in the Pt/g-$C_3N_4$ interface. Similarly, the depletion layer at Pt/BiOI was formed due to the higher work function of BiOI in comparison to Pt. Under visible light irradiation, the inverse electric field at Pt/g-$C_3N_4$ and Pt/BiOI would induce the $e^-$ in BiOI and $h^+$ in g-$C_3N_4$ to combine at the Pt metal. Since this charge transfer process occurred without overcoming the Schottky barrier, it was termed as an unobstructed Z-scheme heterojunction and enabled the photogenerated $e^-$ in the CB of g-$C_3N_4$ and $h^+$ in the VB of BiOI to form $^\bullet O_2^-$ and $^\bullet OH$ radicals that efficiently degraded the organic contaminants [136].

### 5.4. Carbonaceous Materials Compounding

Carbonaceous materials, such as graphene, carbon nanotubes (CNTs), carbon quantum dots (CQDs), carbon fibres, multi-walled carbon nanotubes (MWCNTs), carbon spheres, etc., are reported to play a vital role in enhancing the photocatalytic performance of BiOX/$Bi_xO_yX_z$ nanomaterials [94,145,147,148]. Graphene or reduced graphene oxide (rGO) has been considered a good electron collector and charge transport medium in photocatalysis owing to its high conductivity, excellent electron mobility, and large specific surface area. BiOCl/carbon-based photocatalysts have gained enormous interest due to their enhanced performance, which was attributed to their strong adsorption, excellent light absorption, and rapid transfer of photogenerated charges [228,229]. A BiOCl/CQDs/rGO ternary heterojunction photocatalyst driven by visible light exhibited enhanced ciprofloxacin removal efficiency, which was attributed to the excellent adsorption, enhanced charge separation and charge injection induced by the presence of CQDs and rGO. The photocatalytic efficiency of BiOCl/CQDs/rGO was 3.8 and 10.4 times greater in comparison to BiOCl/CQDs and BiOCl, respectively, while the removal efficiency was ~87% [230]. Yu et al. hydrothermally synthesized 3D BiOBr/rGO heterostructured aerogel using dopamine as both a reducing agent and cross-linker. The rate of the photodegradation of MO (80%) using 3D BiOBr/rGO was much higher compared to RhB (50%) and phenol (35%) under 60 min of visible light irradiation. Strong π-π interaction through the conjugative aromatic structure was attributed to the highly efficient selective adsorption of anionic MO [231]. Similarly, the addition of 1 wt% rGO relative to BiOBr sheets with exposed {001} facets with a core/shell structure exhibited the highest activity for the photodegradation of orange II dye (97% in 90 min) and the removal of acetaminophen (93% in 105 min). The enhanced photocatalytic activity of (1 wt%) rGO/BiOBr was attributed to increased visible light absorption, effective separation, the transportation of photogenerated charge carriers and the formation of a Schottky barrier at the interface between BiOBr and rGO, which enabled the transfer of $e^-$ from the CB of BiOBr to rGO (due to its higher work function) and the internal electric field at the interface. The capability of rGO to store and shuttle $e^-$ enabled the formation of $^\bullet O_2^-$ radicals by reacting with adsorbed $O_2$ molecules, while allowing the $h^+$ to react with OH to form $^\bullet OH$, the two main species responsible for the oxidation of organic contaminants [232]. Z-scheme heterojunction photocatalysts with solid-state electron mediators bridging two semiconductors were proposed for enhancing the performance through the efficient transport and separation of the photogenerated charge carriers. For instance, a

2D/2D Z-scheme heterojunction was constructed between BiOBr and g-$C_3N_4$ using carbon dots as the solid-state electron mediator, and it exhibited enhanced photocatalytic performance in the degradation of ciprofloxacin (~84% in 105 min) and tetracycline (~83% in 60 min) under visible light degradation. Under visible light irradiation, the photogenerated $e^-$ in the CB of BiOBr with low reduction ability and photogenerated $h^+$ in the VB of g-$C_3N_4$ with low oxidation ability are transferred to the carbon dots, while the $e^-$ and $h^+$ with high reduction and oxidation ability produce $^\bullet O_2^-$ and $^\bullet OH$ active species that react with the organic contaminants for their mineralization into $CO_2$ and $H_2O$ [145]. Similarly, the unique electron mediating feature of carbon dots coupled with BiOBr (20 wt%) and g-$C_3N_4$ nanosheets facilitated the improved separation of photogenerated charge carriers for superior performance towards the degradation of organic contaminants (rhodamine B, methylene blue and methyl orange) and the photoreduction of Cr(VI) to Cr(III) under visible light, with $^\bullet O_2^-$ and $^\bullet OH$ being the active species [94]. Likewise, rGO was employed as an electron transfer mediator in the heterojunction formed between BiOBr (10 wt%) with protonated g-$C_3N_4$ for the photodegradation of tetracycline (59% mineralized) and BiOCl with protonated g-$C_3N_4$ for the photodegradation of tetracycline (96% in 180 min) and the selective oxidation of benzyl alcohol (conversation rate 76% and selectivity 99%). In both cases, the mechanism of photocatalysis followed the Z-scheme heterojunction, with $^\bullet O_2^-$ and $^\bullet OH$ being the primary active species [144,233]. On the other hand, MWCNTs were also reported to have been employed as electron mediators in the heterojunction between g-$C_3N_4$ and BiOI (20 wt%). The heterojunction exhibited improved visible light photocatalytic activity towards the degradation of methylene blue (10 ppm, 70% in 3 h) under visible light (>420 nm) through the Z-scheme mediated charge transfer [142]. A p-n junction formed by coupling g-$C_3N_4$ and BiOBr with rGO as the conductive support exhibited enhanced photocatalytic activity in the degradation of rhodamine B (10 ppm, 66% in 60 min) under visible light. The $sp^2$-hybridized carbon atoms in graphene capable of storing and shuttling electrons enabled the photogenerated electrons from the CB of BiOBr to flow into it and formed a Schottky barrier at the interface for preventing their backflow. Meanwhile, the electrons with high reduction potential and holes with high oxidation potential reacted with dissolved $O_2$ and $OH^-$ to form $^\bullet O_2^-$ and $^\bullet OH$ radicals, which were actively involved in the photodegradation of rhodamine B [146]. In comparison to CNTs, carbon dots and rGO, mussel-inspired biometric carbon material polydopamine, also possessing a conjugated π structure and good electron transport ability, has attracted significant interest owing to its excellent adhesion ability, strong light-harvesting capacity, photoconductivity and biocompatibility. The Z-scheme heterojunction photocatalyst g-$C_3N_4$@polydopamine/BiOBr showed high activity in the photocatalytic degradation of sulfamethoxazole under visible light. Polydopamine was reported to promote the efficient separation of the photogenerated charge carriers for ensuring efficient redox capability of the photocatalyst, while the mechanism studied through radical quenching experiments confirmed that the $h^+$ and $^\bullet O_2^-$ were the major reactive species for oxidizing sulfamethoxazole [169].

### 5.5. Integration of Other Semiconductor Nanostructures

Single component photocatalysts fail to exhibit higher photocatalytic efficiency due to the rapid recombination of the photogenerated charge carriers. In order to achieve enhanced photocatalytic efficiency, one of the most common strategies is to construct a heterojunction photocatalytic system by coupling two or more semiconductors [234]. Typically, in a heterojunction photocatalytic system, the photogenerated electrons in the CB of photocatalyst A migrate to the CB of photocatalyst B, while the photogenerated holes in the VB of photocatalyst B move to the VB of photocatalyst A, curbing their recombination due to spatial isolation. However, after the charge transfer, the redox ability of the photogenerated charges becomes weakened since the top of the VB potential of photocatalyst A is less positive than that of photocatalyst B, and the bottom of the CB potential of photocatalyst B is less negative than that of photocatalyst A. Due to this

drawback, the heterojunction photocatalytic system (referred to as type-II heterojunction) fails to simultaneously possess high charge-separation efficiency and strong redox ability. Therefore, a Z-scheme photocatalytic process was proposed by carefully studying the natural photosynthesis reaction in plants, which also features the spatial isolation of the photogenerated charges to hinder their recombination. Although the structure of direct Z-scheme photocatalyst is similar to that of a type-II heterojunction photocatalyst, its charge-carrier migration mechanism is different and the pathway resembles the letter "Z". During the photocatalytic reaction, the photogenerated electrons in photocatalyst B with lower reduction ability recombine with the photogenerated holes in photocatalyst A with lower oxidation ability. Therefore, the photogenerated electrons in photocatalyst A with high reduction ability and the photogenerated holes in photocatalyst B with high oxidation ability can perform the redox reactions without any hindrance, and the performance of the resulting Z-scheme photocatalytic system can be optimized. Further, the large number of defects aggregated at the contact interface exhibits properties similar to that of conductors with low electrical resistance owing to the fact that energy levels at the interface become quasi-continuous [235,236].

Since 2D nanostructures can offer an apt platform for establishing surface contact with other species, the idea of constructing a heterojunction by the hybridization of two types of 2D photocatalysts is an appropriate strategy for increasing the interface area. Recently, the combination of g-$C_3N_4$ with BiOX/$Bi_xO_yX_z$ for the construction of 2D/2D heterojunction photocatalysts has attracted considerable research attention [237–240]. Liu et al. reported the fabrication of a BiOBr/g-$C_3N_4$ heterojunction through a simple reflux process, and its photocatalytic performance was studied by the degradation of rhodamine B and bisphenol A under visible light irradiation. The enhanced photocatalytic performance of the heterojunction composite was obviously attributed to the efficient charge generation and separation, while the active species involved during the photodegradation of rhodamine B and bisphenol A were found to be in the order $^\bullet OH > h^+ > {}^\bullet O_2^-$. For determining whether the photocatalytic mechanism followed the type-II heterojunction or Z-scheme system, the migration channel of the photogenerated electron-hole pairs was analysed through UV-Vis diffused reflectance spectroscopy and X-ray photoelectron spectroscopy. The results revealed that the values of CB and VB potentials of pristine BiOBr nanoplates were 0.30 eV and 3.07 eV, while those of pristine g-$C_3N_4$ nanosheets were $-1.12$ eV and 1.58 eV, respectively. After the hybridization of g-$C_3N_4$ with BiOBr, the VB edge of the BiOBr/g-$C_3N_4$ heterojunction was found to be shifted to 1.32 eV due to the alignment of the Fermi levels at the interface. Under visible light irradiation, the photogenerated electrons moved from the CB of g-$C_3N_4$ to that of BiOBr, while the photogenerated holes moved from the VB of BiOBr to that of g-$C_3N_4$ across the intimate well-aligned band structure due to the potential difference. As observed from Figure 12a, if BiOBr/g-$C_3N_4$ had formed a type-II heterojunction, the formation of $^\bullet O_2^-$ and $^\bullet OH$ would not have been possible due to the insufficient reduction and oxidation potential of BiOBr and g-$C_3N_4$. Therefore, the Z-scheme photocatalytic system was found to be constructed as shown in Figure 12b, wherein the electrons accumulated in the CB of g-$C_3N_4$ ($-1.12$ eV vs. NHE) reacted with oxygen molecules to form $^\bullet O_2^-$, and the holes in the VB of BiOBr (3.07 eV vs. NHE) reacted with $OH^-$ to generate $^\bullet OH$ radicals [123].

A direct solid-state Z-scheme heterojunction photocatalyst was constructed by coupling nanosheets of BiOI and g-$C_3N_4$ for the photodegradation of toxic microcystin-LR under visible light irradiation. The rate constant of the best performing g-$C_3N_4$/BiOI heterojunction photocatalyst (0.4357 h$^{-1}$) was three and five times greater than pristine BiOI and g-$C_3N_4$, respectively, and radical scavenger studies revealed that $^\bullet O_2^-$ played the major role in the degradation of microcystin-LR. If BiOI/g-$C_3N_4$ had formed a type-II heterojunction, the formation of $^\bullet OH$ and $^\bullet O_2^-$ would not have been possible due to the insufficient reduction and oxidation potential of BiOI and g-$C_3N_4$. Therefore, the direct Z-scheme charge transfer mechanism occurred, wherein the photogenerated charges formed in the CB of g-$C_3N_4$ and the VB of BiOI with high reduction and oxidation ability reacted

with $O_2$ and $OH^-$ to generate $\bullet O_2^-$ and $\bullet OH$, respectively, while the photogenerated charges with low reduction and oxidation ability recombined at the interface [111]. Tian et al. reported the fabrication of two p-n junction photocatalysts by coupling different facets of BiOI with g-$C_3N_4$ through a simple precipitation method, and studied their feasibility for the photodegradation of various organic contaminants such as 2,4-dichlorophenol, bisphenol A, rhodamine B and tetracycline hydrochloride [112]. Typically, the {001} facet of BiOI was coupled with the {002} facet of g-$C_3N_4$ to form a (001)-BiOI/(002)-g-$C_3N_4$ photocatalyst through parallel assembly, and (110)-BiOI/(002)$^+$-g-$C_3N_4$ was fabricated by the vertical assembly of the {110} facet of BiOI on the positively charged {002} facet of g-$C_3N_4$. The results indicated that the top-top facets coupled (001)-BiOI/(002)-g-$C_3N_4$ photocatalyst exhibited more than four times enhanced performance in the photodegradation of bisphenol A and tetracycline hydrochloride in comparison to the laterally assembled (110)-BiOI-(002)$^+$-g-$C_3N_4$.

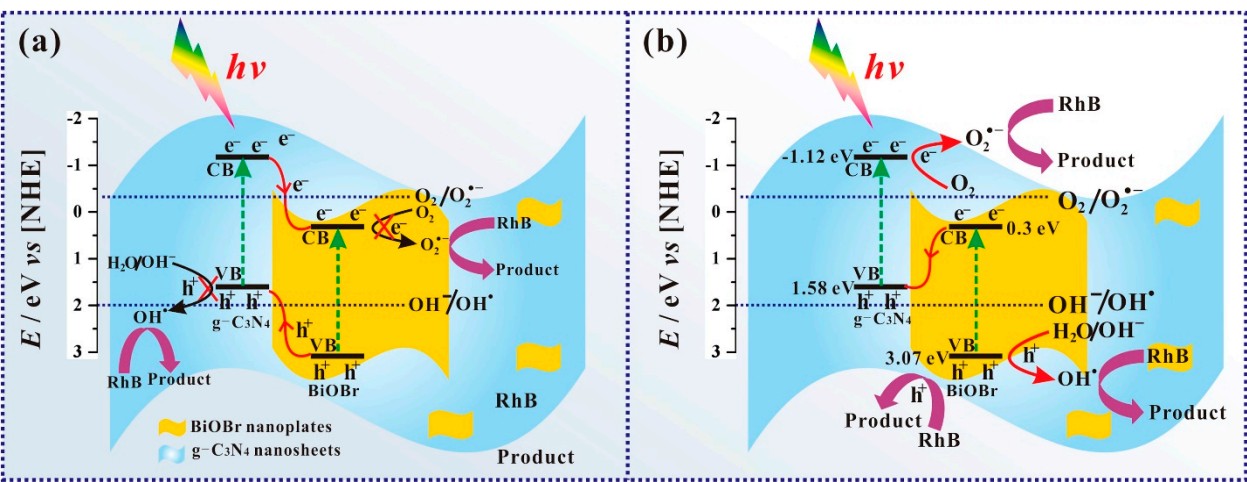

**Figure 12.** Schematic depicting (**a**) type-II heterojunction and (**b**) Z-scheme system, the two possible photocatalytic charge transfer processes in the heterojunction between BiOBr nanoplates and g-$C_3N_4$ nanosheets under visible light irradiation. Reprinted from Ref. [123] with permission from Elsevier.

As shown in Figure 13a, the fermi energy level of BiOI as a p-type semiconductor is located close to the VB, while in the case of g-$C_3N_4$, it is located close to the CB and the energy levels of both the semiconductors achieve an equilibrium to form a (001)-BiOI/(002)-g-$C_3N_4$ p-n heterojunction photocatalyst. The formation of the p-n junction effectively separates the photogenerated electron-hole pairs in both the heterojunction photocatalysts, but the transfer rate of the photogenerated electrons was found to be distinctly different in the two heterojunctions. In the case of the (001)-BiOI/(002)-g-$C_3N_4$ p-n heterojunction photocatalyst shown in Figure 13b, the IEF of BiOI along the [1] direction lying perpendicular to the g-$C_3N_4$ nanosheets results in the rapid enrichment of the electrons on g-$C_3N_4$ that benefitted the subsequent reduction reactions for the generation of $^1O_2$ and $\bullet O_2^-$ radical species. On the other hand, in the case of the (110)-BiOI/(002)$^+$-g-$C_3N_4$ p-n heterojunction photocatalyst shown in Figure 13c, the charge transfer direction was parallel, and due to the long diffusion distance, some of the electrons recombined with the holes, leading to inefficient charge transfer in the heterojunction.

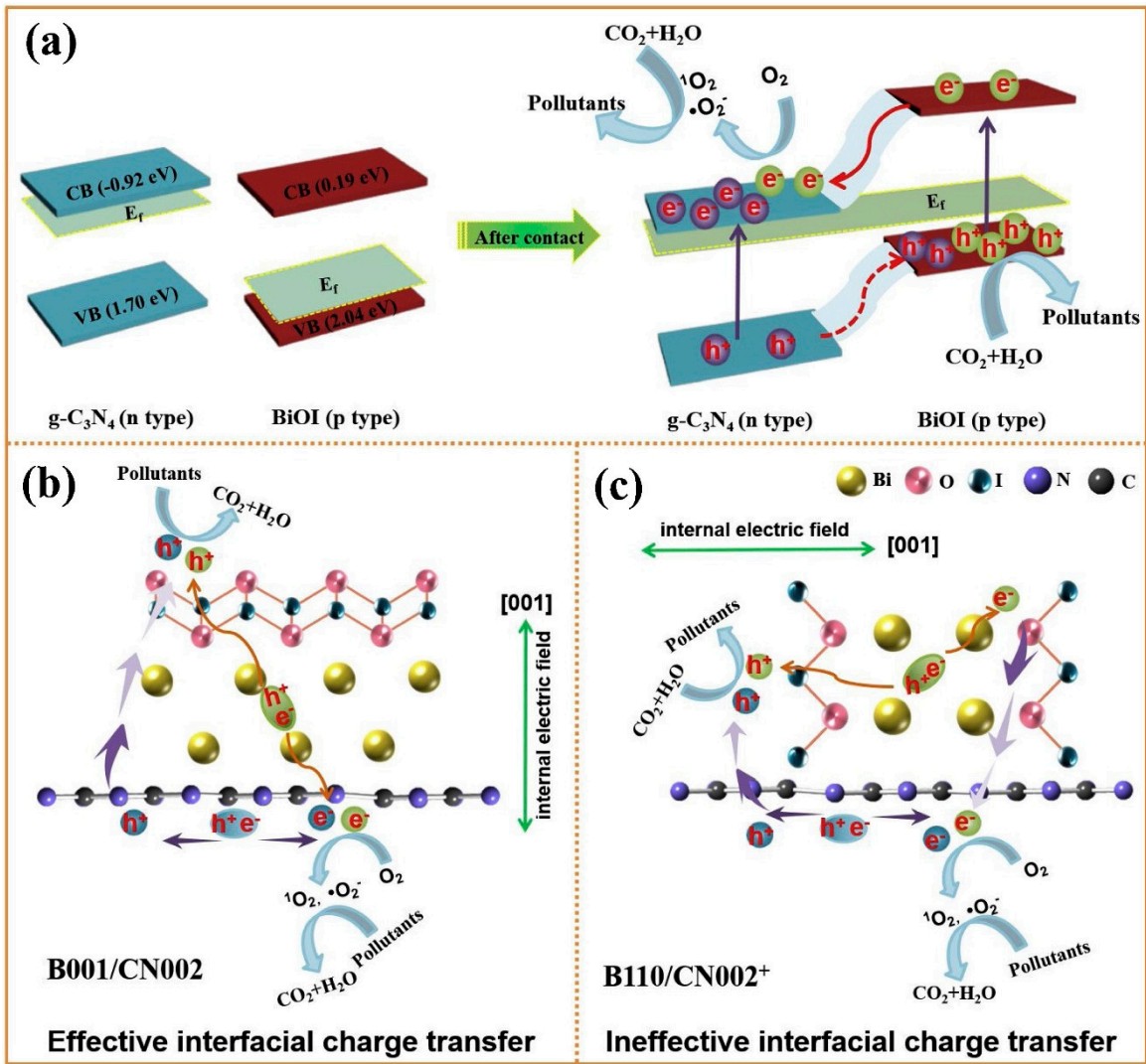

**Figure 13.** (**a**) Schematic depicting the formation of p-n junction and the proposed charge separation process in a heterojunction formed by coupling the top-top facets of BiOI and g-$C_3N_4$, i.e., (001)-BiOI/(002)-g-$C_3N_4$ under visible light irradiation. Schematic depicting the proposed photodegradation mechanism over (**b**) (001)-BiOI/(002)-g-$C_3N_4$ and (**c**) (110)-BiOI/(002)$^+$-g-$C_3N_4$ (positively charged g-$C_3N_4$). Reprinted from Ref. [112] with permission from Elsevier.

The development of ternary or multicomponent heterojunction systems was considered owing to the possibility of enhancing the charge separation and transfer ability and extending the scope of light absorption as compared to binary systems. For instance, the AgBr@g-$C_3N_4$/BiOBr ternary composite was fabricated through hydrothermal processing and an in situ ion-exchange route for dispersing AgBr nanoparticles between the g-$C_3N_4$/BiOBr (2D/2D) heterojunction. Interestingly, BiOBr played a central role between g-$C_3N_4$ and AgBr for providing a high-speed charge transfer channel and isolating the photogenerated charge carriers, resulting in high photocatalytic efficiency for the degradation of rhodamine B (10 ppm, 94% in 30 min) and tetracycline hydrochloride (10 ppm, 78% in 2 h) [160]. The ternary heterojunction between $Bi_{24}O_{31}Cl_{10}$, $MoS_2$ and g-$C_3N_4$ was synthesized through the impregnation-calcination method. The higher photocatalytic efficiency of the g-$C_3N_4$/$MoS_2$/$Bi_{24}O_{31}Cl_{10}$ ternary heterojunction photocatalyst in the degradation of tetracycline hydrochloride (20 ppm, ~97% in 50 min) under visible light was attributed to its enhanced light absorption capacity, the rapid separation of the photogenerated charges and the strong redox ability. The mechanism of charge transfer was reported to follow a dual Z-scheme pathway as depicted in Figure 14, wherein the photogenerated e$^-$ with less reduction ability from the CB of g-$C_3N_4$ and $MoS_2$ jump to the VB of $Bi_{24}O_{31}Cl_{10}$ for

recombining with the holes, while the e$^-$ with strong reduction ability and h$^+$ with strong oxidation stability are spared. Scavenger studies and electron spin resonance spectroscopy confirmed the involvement of $\bullet$O$_2^-$ and $\bullet$OH radical species, which also confirms the transfer of the photogenerated charge carriers through the dual Z-scheme pathway for ensuring enhanced photocatalytic efficiency [155].

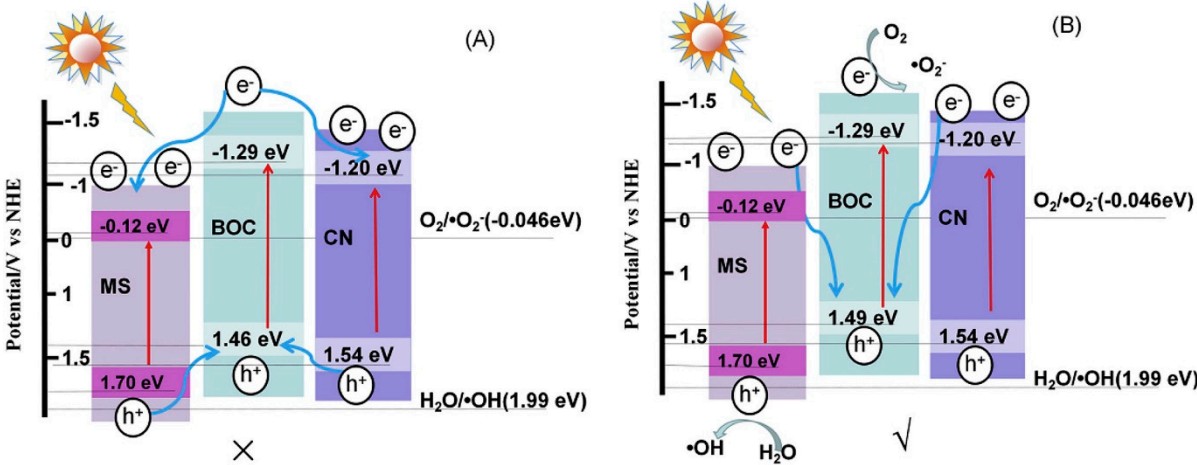

**Figure 14.** Schematic representation of the possible steps involved during the photocatalytic process and the charge transfer mechanism in the g-C$_3$N$_4$/MoS$_2$/Bi$_{24}$O$_{31}$Cl$_{10}$ composite: (**A**) traditional pathway and (**B**) dual Z-scheme pathway. Reprinted from Ref. [155] with permission from Elsevier.

Recent reports on nanostructured heterojunctions of g-C$_3$N$_4$/BiOI show their poor dispersion in water, and they easily aggregate because of their higher surface energy. This leads to a remarkable reduction in their photocatalytic activity. In contrast, 1D nanofibres with a high surface area and high aspect ratios have potential in overcoming these problems. In particular, the 3D macroscopic structure of electrospun polyacrylonitrile nanofibres with excellent hydrophobicity can minimize agglomeration and improve the separation of the nanostructured heterojunctions of g-C$_3$N$_4$/BiOI in water for practical applications [150]. The fabrication of sandwich-like BiOI/AgI/g-C$_3$N$_4$ through the in situ crystallization approach showed good photocatalytic performance in degrading MO and the reduction of Cr(VI) ions under visible light irradiation. The AgI in the composite served as a charge transmission bridge between BiOI and g-C$_3$N$_4$ that resulted in more efficient charge transfer and better separation of charge carriers [152]. Jiang et al. developed a novel ternary BiOI/g-C$_3$N$_4$/CeO$_2$ photocatalyst through calcination and hydrothermal treatment. This composite exhibited superior photocatalytic performance, which was far higher than that of either the single component or two component systems. For this photocatalytic BiOI/g-C$_3$N$_4$/CeO$_2$ heterojunction system, 91.6% of tetracycline was degraded in 120 min, owing to the double charge transfer process between the g-C$_3$N$_4$ and the other catalysts in the ternary heterojunction and the enhanced separation efficiency of photogenerated electron-hole pairs [153].

### 5.6. Coupling BiOX and BiOY with g-C$_3$N$_4$

The approach of coupling two semiconductors to form a layered structure with an interfacial electric field is particularly promising since this enhances the possibility of satisfying the band alignment requirements for water splitting through the band structure, and subsequently boosts the separation between the photogenerated electron-hole pairs. In this context, heterolayers of BiOX$_1$/BiOX$_2$ (with X$_1$ and X$_2$ being different halides) are plausibly superior in comparison to homogeneous BiOX bilayers owing to the possibility of heterojunction induced separation of photogenerated electron-hole pairs [241]. For instance, a ternary heterojunction between BiOI, BiOCl and g-C$_3$N$_4$ with different weight ratios was fabricated through the precipitation technique, among which BiOI(50)-BiOCl(30)/g-

$C_3N_4(20)$ exhibited enhanced photodegradation of acid orange 7 (10 ppm, 97% in 140 min) under visible light irradiation in comparison to pristine and other binary/ternary heterojunction counterparts. Under visible light exposure, the photogenerated $e^-$ in the CB of $g\text{-}C_3N_4$ with low reduction ability were transferred to the CB of BiOCl and BiOI, while the $h^+$ with low oxidation ability were transferred from the VB of BiOI to the VB of $g\text{-}C_3N_4$, resulting in efficient charge separation and enabling the $e^-$ and $h^+$ with higher reduction and oxidation capabilities to participate in the photodegradation of acid orange 7 [91]. Similarly, a ternary composite of $g\text{-}C_3N_4/BiOI/BiOBr$ synthesized through the hydrothermal approach exhibited enhanced performance in the photodegradation of methylene blue (20 ppm, 80% in 150 min) under visible light irradiation. Matching band positions of $g\text{-}C_3N_4$, BiOI and BiOBr allowed the transfer of charge carriers through a direct Z-scheme that favoured the efficient separation and transfer of the photogenerated charge carriers for the effective generation of reactive oxygen species [167]. The deposition-precipitation process was reported for synthesizing 2D $g\text{-}C_3N_4@BiOCl/Bi_{12}O_{17}Cl_2$ composites that were employed in the removal of nitric oxide with fixed concentration mixed with air stream under ambient temperature in a continuous flow reactor. The nitric oxide removal efficiency using the $g\text{-}C_3N_4@BiOCl/Bi_{12}O_{17}Cl_2$ heterojunction photocatalyst (46.8% in 30 min) was found to be greater than pristine $BiOCl/Bi_{12}O_{17}Cl_2$ (36.2%) and $g\text{-}C_3N_4$ (14.6%) under visible light irradiation, and was attributed to the intimate contact interface, suitable band structure, larger pore volume and improved visible light absorption [165]. Chou et al. reported the fabrication of various types of $BiO_xI_y/g\text{-}C_3N_4$ heterojunction composites through the hydrothermal method towards the photodegradation of crystal violet. Interestingly, $Bi_7O_9I_3/Bi_5O_7I/g\text{-}C_3N_4$ was found to exhibit superior performance in comparison to $BiOI/g\text{-}C_3N_4$, $Bi_7O_9I_3/g\text{-}C_3N_4$ and $Bi_5O_7I/g\text{-}C_3N_4$ under visible light irradiation, which was ascribed primarily to the formation of a synergistic ternary heterojunction that ensured the separation of photogenerated charge carriers [163].

## 6. Conclusions and Future Perspectives

Recent years have witnessed significant progress in visible light driven photocatalysis aided by the comprehensive understanding of the structure-to-property relationship of nanostructured materials. Progress on molecularly thin 2D nanosheets has been phenomenal since the discovery of graphene, and studies in the past decade explored their customizable ultrathin architecture, composition and functionality driven by the exceptional physical, chemical, optical and electronic properties arising due to the unique ability of the nanosheets to confine electrons. Advancement towards 2D/2D heterojunction photocatalysts originated as a solution for tackling the rapid recombination of the photogenerated charge carriers in single component systems. However, many interesting studies pertaining to various 2D nanostructured photocatalysts are being pursued, and herein we have presented a comprehensive overview on the recent advances in the design, preparation, and photocatalytic applications of $BiOX/Bi_xO_yX_z\text{-}g\text{-}C_3N_4$ heterojunction photocatalysts. The band structure of the individual components, the resulting properties and plausible outcomes during heterojunction formation were summarized. Then, various methods for fabricating $BiOX/Bi_xO_yX_z\text{-}g\text{-}C_3N_4$ heterojunction photocatalysts were thoroughly discussed, emphasizing the dimensional anisotropy and morphological evolution that led to enhanced performance. Applications of the $BiOX/Bi_xO_yX_z\text{-}g\text{-}C_3N_4$ photocatalysts in the degradation of various organic contaminants, $H_2$ generation, $CO_2$ reduction, $N_2$ fixation and organic synthesis were summarized. Further, the improvement in the performance of $BiOX/Bi_xO_yX_z\text{-}g\text{-}C_3N_4$ due to defects, facets and by the integration of metals, semiconductors and carbon materials is emphasized. The formation of the type-II heterojunction and Z-scheme bridge complimented with their structural stability is specified at relevant sections, and several salient studies are featured to stimulate the desire of the researchers to find a breakthrough.

Several studies were reported on the usage of $BiOX/Bi_xO_yX_z\text{-}g\text{-}C_3N_4$ heterojunction photocatalysts for organic contaminant degradation, and some reports were available on the

photocatalytic reduction of $CO_2$ as summarized in Tables 1 and 2. Despite the encouraging results on photocatalytic $H_2$ generation, $O_2$ evolution and $N_2$ fixation using $Bi_xO_yX_z$ and g-$C_3N_4$, it was surprising that no studies were reported to date on BiOX/$Bi_xO_yX_z$-g-$C_3N_4$ heterojunction photocatalysts. Therefore, plenty of opportunity exists for designing efficient heterojunction photocatalysts by strategically coupling engineered $Bi_xO_yX_z$ and g-$C_3N_4$, as detailed below.

(1) Many recent studies reported the synthesis of defect-rich g-$C_3N_4$ for realizing enhanced activity for $H_2$ generation [242–245]. The introduction of defects in the form of nitrogen vacancies in g-$C_3N_4$ induced the formation of midgap states under the CB that resulted in the extension of the visible light absorption, and trapped the photogenerated $e^-$ to minimize recombination loss while facilitating its rapid transfer. Forming a heterojunction by combining defect-rich g-$C_3N_4$ with defect-rich BiOX/$Bi_xO_yX_z$ can enhance $H_2$ generation.

(2) The inherent drawback of g-$C_3N_4$ has been its poor mass diffusion and charge separation efficiency for achieving enhanced photocatalytic $O_2$ evolution efficiency. Modulating the band structures of g-$C_3N_4$ (by protonation or the addition of defects and dopants) was reported to enhance its efficiency [246], and therefore a heterojunction photocatalyst constructed between $Bi_xO_yX_z$ with exposed facets and band structure modulated g-$C_3N_4$ could achieve enhanced quantum efficiencies.

(3) Despite the CB of g-$C_3N_4$ being more negative than $N_2$/$NH_3$ reduction potential, its low conductivity and high recombination rate are some of the impediments that deter its potential for photocatalytic $N_2$ fixation. The concurrent addition of dopants and defects (carbon/nitrogen vacancies) was reported to improve its photocatalytic $N_2$ fixation efficiency [247,248]. Heterojunction photocatalysts constructed between doped BiOX/$Bi_xO_yX_z$ with exposed facets and doped/defect-rich g-$C_3N_4$ are expected to exhibit enhanced photocatalytic $N_2$ fixation efficiency.

(4) Another interesting opportunity is the construction of a heterojunction between an atomically thin layer of $Bi_xO_yX_z$ and g-$C_3N_4$, which can be very challenging. However, the unique physical and chemical properties in addition to the easy formation of surface defects could pave the way towards enhanced quantum efficiencies.

**Funding:** This research received no external funding.

**Acknowledgments:** This work was supported by the Department of Science and Technology, Government of India through the DST INSPIRE Faculty project (No. IFA15 MS-41) and was funded by the MEXT Promotion of Distinctive Joint Research Center Program Grant Number JPMXP 0618217662. SP acknowledges the European Regional Development Grant for providing Ser Cymru-II Rising Star Fellowship through the Welsh Government (80761-SU-102-West).

**Conflicts of Interest:** The authors declare no conflict of interest.

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
