# Peer review of "Advanced Two-Dimensional Heterojunction Photocatalysts of Stoichiometric and Non-Stoichiometric Bismuth Oxyhalides with Graphitic Carbon Nitride for Sustainable Energy and Environmental Applications"

_catalysts, doi:10.3390/catal11040426_

Round 1
Reviewer 1 Report
Reviewer’s Report
Advanced two-dimensional heterojunction photocatalysts of stoichiometric and non-stoichiometric bismuth oxyhalides with graphitic carbon nitride for sustainable energy and environmental applications
Manuscript ID: catalysts-1158082
This article has reviewed the recent progress on the 2D/2D hetero-junction constructed between BiOX/BixOyXz with graphitic carbon nitride (g-C3N4). This review also emphasized the band structure of individual components, various fabrication methods, different strategies developed for improving the photocatalytic performance and their applications in the degradation of various organic contaminants, hydrogen (H2) evolution, carbon dioxide (CO2) reduction, nitrogen (N2) fixation and organic synthesis of clean chemicals. On the whole, the review is well organized and it covers the vast literature on the BiOX/BixOyXz-g-C3N4 heterojunction photocatalysts. Therefore, I recommend the publication of this review in the journal with minor changes.
- Some figures have a poor quality that should be improved.
- Some sections of the manuscript are not properly connected.
- It seems that the generality of the conclusion section is weak.
- Some typos could be found. The authors should correct those errors before publication.
Author Response
Comments from Reviewer 1:
This article has reviewed the recent progress on the 2D/2D hetero-junction constructed between BiOX/BixOyXz with graphitic carbon nitride (g-C3N4). This review also emphasized the band structure of individual components, various fabrication methods, different strategies developed for improving the photocatalytic performance and their applications in the degradation of various organic contaminants, hydrogen (H2) evolution, carbon dioxide (CO2) reduction, nitrogen (N2) fixation and organic synthesis of clean chemicals. On the whole, the review is well organized and it covers the vast literature on the BiOX/BixOyXz-g-C3N4 heterojunction photocatalysts. Therefore, I recommend the publication of this review in the journal with minor changes.
We thank the reviewer the time dedicated for thoroughly reading the manuscript and for giving constructive comments for improving our article. Based on these, we have revised the manuscript.
1. Some figures have a poor quality that should be improved.
Author’s reply: We thank the reviewer for this suggestion. In accordance to the editorial comment, we have now incorporated high-resolution images in the revised manuscript corresponding to Figure 7, 8, 10, 11, 12, 13 and 14 that were downloaded from the cited articles.
2. Some sections of the manuscript are not properly connected.
Author’s reply: We have tried our best to connect the sections as effectively as possible. Since we have tried to incorporate as much information as possible it appears a little disconnected. However, as mentioned in the conclusion part lines 1285-1288, we wish that our review could stimulate the desire of the researchers to find a breakthrough.
3. It seems that the generality of the conclusion section is weak.
Author’s reply: We have made changes in lines 1296, 1297 and 1319 to make the conclusion more appealing.
4. Some typos could be found. The authors should correct those errors before publication.
Author’s reply: As per the editorial comment, we have checked for errors throughout the manuscript and have corrected them.
Reviewer 2 Report
This authors systematically summarized the recent progress on the 2D/2D heterojunction constructed between BiOX/BixOyXz with graphitic carbon nitride (g-C3N4). The review presented a summary on the band structure of BixOyXz and furnished information on the various methods of coupling BiOX/BixOyX and g-C3N4 to fabricate heterojunction photocatalysts for organic contaminant degradation, H2 generation, CO2 reduction, N2 fixation and organic synthesis applications. At the same time, the various strategies for improving the performance of g-C3N4-BiOX/BixOyXz heterojunction photocatalysts viz., creation of defects, role of facets, integration with other semiconductors, metals and carbon materials are discussed. Also, the future prospects of BiOX/BixOyXz-g-C3N4 heterojunction photocatalysts for broader energy and environmental applications are deliberated.
This paper is well organized and It includes sufficient information about the two-dimensional heterojunction photocatalysts constructed between BiOX/BixOyXz with graphitic carbon nitride (g-C3N4). However, There are a few layout mistakes, for exsmple, line 506, Table 1.I recommend publication after minor revision.

Author Response
Comments from Reviewer 2:
This authors systematically summarized the recent progress on the 2D/2D heterojunction constructed between BiOX/BixOyXz with graphitic carbon nitride (g-C3N4). The review presented a summary on the band structure of BixOyXz and furnished information on the various methods of coupling BiOX/BixOyX and g-C3N4 to fabricate heterojunction photocatalysts for organic contaminant degradation, H2 generation, CO2 reduction, N2 fixation and organic synthesis applications. At the same time, the various strategies for improving the performance of g-C3N4-BiOX/BixOyXz heterojunction photocatalysts viz., creation of defects, role of facets, integration with other semiconductors, metals and carbon materials are discussed. Also, the future prospects of BiOX/BixOyXz-g-C3N4 heterojunction photocatalysts for broader energy and environmental applications are deliberated.
This paper is well organized and It includes sufficient information about the two-dimensional heterojunction photocatalysts constructed between BiOX/BixOyXz with graphitic carbon nitride (g-C3N4). However, there are a few layout mistakes, for example, line 506, Table 1. I recommend publication after minor revision.
Author’s reply: We thank the reviewer for the time spent on thoroughly reading our manuscript and constructive comments its improvement. As per the editorial comment, we have corrected the layout mistake in Table 1 in the revised manuscript and have made sure that there are no other mistakes.